



# Interannual variability in the Australian carbon cycle over 2015-2019, based on assimilation of OCO-2 satellite data

Yohanna Villalobos[1,2,5], Peter J. Rayner[1,2,3], Jeremy D. Silver[1,4], Steven Thomas[1], Vanessa Haverd[5,†], Jürgen Knauer[9,5], Zoë M. Loh[6], Nicholas M. Deutscher[7], David W.T. Griffith[7], and David F. Pollard[8]

[1]School of Geography, Earth and Atmospheric Sciences, University of Melbourne, Australia
[2]ARC Centre of Excellence for Climate Extremes, Sydney, Australia
[3]Climate & Energy College, University of Melbourne, Melbourne, Australia
[4]School of Mathematics and Statistics, University of Melbourne, Melbourne, Australia
[5]CSIRO Oceans and Atmosphere, Canberra, 2601, Australia
[6]CSIRO Oceans and Atmosphere, Aspendale, Victoria 3195, Australia
[7]Centre for Atmospheric Chemistry, School of Chemistry, University of Wollongong, Wollongong, NSW, 2522, Australia.
[8]National Institute of Water and Atmospheric Research Ltd (NIWA), Lauder, New Zealand.
[9]Hawkesbury Institute for the Environment, Western Sydney University, Penrith, NSW 2751, Australia
[†]Deceased, 19 January 2021.
**Correspondence:** Yohanna Villalobos (yohanna.villaloboscortes@csiro.au)

**Abstract.**

In this study, we employ a regional inverse modelling approach to estimate monthly carbon fluxes over the Australian continent for 2015–2019 using the assimilation of the total column-averaged mole fractions of carbon dioxide from the Orbiting Carbon Observatory-2 (OCO-2, version 9). Subsequently, we study the carbon cycle variations and relate their fluctuations to anomalies in vegetation productivity and climate drivers. Our five-year regional carbon flux inversion suggests that Australia was a carbon sink averaging -0.46 ± 0.08 PgC yr$^{-1}$ (excluding fossil fuel emissions), largely influenced by a strong carbon uptake (-1.04 PgC yr$^{-1}$) recorded in 2016. Australia semi-arid ecosystems, such as sparsely vegetated regions (in central Australia) and savanna (in northern Australia), were the main contributors to the carbon uptake in 2016. These regions showed relatively high vegetation productivity, high rainfall and low temperature in 2016. In contrast to the large carbon sink found in 2016, the large carbon outgassing recorded in 2019 coincides with an unprecedented deficit of rainfall and higher than average temperature across Australia. Comparison of the posterior column average $CO_2$ concentration against the Total Carbon Column Observing Networks (TCCON) and in situ measurements offers limited insight into the fluxes assimilated with OCO-2. However, the lack of these monitoring stations across Australia, mainly over ecosystems such as the savanna and areas with sparse vegetation, impedes us from providing strong conclusions. Comparison of our flux inversion to the ensemble mean carbon flux of the OCO-2 Multi-model Intercomparison Project (MIP) (2015-2018) agrees with our findings, and their results also suggest that Australia was a strong carbon sink in 2016 (-0.73 ± 0.41 PgC yr$^{-1}$). The analysis of the variability of the nine models that participate in the OCO-2 MIP also aligns with our findings, and it gives us the confidence to say that changes in rainfall and temperature drive most of the carbon flux variability across Australia.





## 1   Introduction

On average, each year, the global terrestrial biosphere absorbs about one-quarter of total global fossil-fuel $CO_2$ emissions that human activities add to the atmosphere (Friedlingstein et al., 2020). Carbon uptake by the terrestrial biosphere plays an important role in the Earth's carbon cycle and in future climate projections, since they can slow down the rise in atmospheric $CO_2$ concentrations. Due to uncertainties in quantifying carbon fluxes by terrestrial biosphere models (Sitch et al., 2013, 2015), scientists are unsure whether the growth rate of emissions in the atmosphere is going to increase or decrease in the
future. In particular the contributions of semi-arid regional ecosystems such as Australia are uncertain and subject to high variability (Trudinger et al., 2016). Understanding which are the main drivers behind carbon flux variability in semi-arid ecosystems is crucial not only for understanding the global carbon cycle but also for predicting future trends in atmospheric $CO_2$ concentration and consequently the future of climate change.

Australia's contribution to inter-annual global carbon cycle variability has been a topic of interest to the carbon-cycle research
community due to an unusually large land carbon sink anomaly of about -0.70 PgC yr$^{-1}$ (relative to the 2003–2012) recorded in 2011, which alone accounted for 57% of the global terrestrial carbon uptake anomaly in this period. Poulter et al. (2014) suggests that the reason for this large carbon uptake in Australia was due to an increase of vegetation cover as a result of increased precipitation in 2011, one of the wettest years on record for Australia. Another study performed by Trudinger et al. (2016) found similar results to Poulter et al. (2014); they estimated a carbon uptake anomaly of -0.40 to -0.61 PgC yr$^{-1}$ (relative
to the 1982–2013). Global atmospheric inversions based on atmospheric $CO_2$ concentrations also support this unexpected large sink over Australia. A study carried out by Detmers et al. (2015) based on the assimilation of the Greenhouse Gases Observing Satellite (GOSAT) retrievals found that the carbon sink anomaly in Australia in 2011 was about -0.23 PgC yr$^{-1}$ (relative to the period June 2009 – June 2013). All these studies agree that the main driver behind the carbon sink anomaly in 2011 was an increase in the gross primary productivity (GPP) arising from an increase in rainfall which coincides with La Niña event
that occurred from 2010 to 2011. Haverd et al. (2016) suggested that the carbon sink anomaly recorded in 2011 was 90% attributable to a higher than expected carbon uptake by semi-arid ecosystem such as savanna and sparsely vegetated regions, mostly driven by a positive response of these ecoregions to precipitation anomalies.

Ma et al. (2016) suggested that the size of the 2011 carbon sink anomaly in Australia was abruptly reduced in 2012, and was nearly eliminated in 2013 (0.08 PgC yr$^{-1}$) due to a decrease in rainfall across Australia. In this study, the authors show
that Australia's semi-arid ecosystems' productivity is strongly influenced by drivers such as rainfall and temperature. A recent continental-scale inverse modelling study, utilising OCO-2 satellite data, suggests that Australia was a sink of $CO_2$ of -0.41 $\pm$ 0.08 PgC yr$^{-1}$ for 2015 (Villalobos et al., 2021). In this study the authors indicate that the stronger carbon sink estimated in 2015 was primarily driven by an increase in productivity over the savanna and sparsely vegetated regions. In this study, the authors also mentioned that periods with a stronger carbon uptake were likely related to increased rainfall in Australian
semi-arid ecosystems.

The current study builds upon the work of Villalobos et al. (2021), which only performed an inversion for 2015 and focused on the total mean of carbon flux for that period. In this study, we assimilate the total column average retrieval from NASA's





Orbiting Carbon Observatory−2 (OCO-2) to study the interannual variability of the Australia carbon fluxes for the period
2015–2019. An interesting question is whether the large carbon sink estimated in 2015 over semi-arid ecosystems will follow
the same patterns after this year or whether such patterns will become stronger or weaker due to changes in precipitation and
temperature. Our paper is organized as follows: Sect. 2 describes the methodology and data we used to perform the inversion,
which also includes a description of the climate drivers and auxiliary data. Section 3 presents the results of the five-year
inversion, analysis of the long-term mean, and variability of the Australian carbon fluxes. In this section, we also show an
analysis of inter-annual variability of rainfall and temperature and the Enhanced Vegetation Index (EVI) variability as a proxy
of the vegetation productivity in Australia. Section 4 presents some discussion about the climatological seasonal difference
between the prior and posterior flux estimates, GPP variability, and a comparison of our assimilated fluxes to the OCO-2 MIP
(2015–2018). Finally, in Sect. 5, we summarize the results of this study.

## 2   Method and data

We follow the same four-dimensional variational data assimilation approach used to estimate the Australia carbon fluxes
described in Villalobos et al. (2021). In this section we will give a brief description of the system and the data used in the
inversion. Further details can be found in Villalobos et al. (2020, 2021).

### 2.1   Inversion set-up

Our regional inversion system optimizes monthly-mean gridded-based surface carbon emissions $x$ using a four-dimensional
variational data assimilation method, which was configured to use the Community Multi-scale Air Quality Model (CMAQ)
(version, v5.3) and its adjoint (version 4.5.1; Hakami et al., 2007). Each year of the five-year period was run independently,
with a spin up of one month for each year. Our system optimizes $CO_2$ surface fluxes by finding the minimum of the cost
function $J(x)$ shown in Eq. 1. Notation in this study follows Rayner et al. (2019).

$$J(\boldsymbol{x}) = \frac{1}{2}\left[(\boldsymbol{x}-\boldsymbol{x^b})^T \mathbf{B}^{-1}(\boldsymbol{x}-\boldsymbol{x^b})\right] + \frac{1}{2}\left[(\mathbf{H}(\boldsymbol{x})-\boldsymbol{y})^T \mathbf{R}^{-1}(\mathbf{H}(\boldsymbol{x})-\boldsymbol{y})\right] \tag{1}$$

This cost function measures the mismatch between a the CMAQ forward model simulation $\mathbf{H}$ and OCO-2 satellite observa-
tions $\boldsymbol{y}$ and the deviation of the control vector $\boldsymbol{x}$ from its background (also termed prior) estimate $\boldsymbol{x^b}$. In our case, the control
vector $\boldsymbol{x}$ (vector of unknowns) consists not only of the gridded $CO_2$ surface fluxes, but also incorporates initial and boundary
conditions. These two latter variables were incorporated into the control vector to reduce any potential biases related to the
boundary inflow that could affect our system (information of how we treat the boundary and initial condition in our system
can be found in Sect. 2.2 in Villalobos et al., 2020, 2021). $\mathbf{R}$ represents the observational error covariance matrix, which was
defined as a diagonal matrix (full description of this covariance matrix is found in Sect.2.3 in Villalobos et al., 2020) and a
brief explanation of how was constructed is found in Sect. 2.3. $\mathbf{B}$ is the associated error covariance matrix of $\boldsymbol{x_b}$, boundary and
initial concentrations, and includes off-diagonal terms. In these off-diagonal values, we only include spatial and non-temporal



correlations of the prior fluxes (details of the structure of the prior error covariance matrix is found in Section 2.4 in Villalobos et al., 2020), and summary of its description is found in Sect 2.2.

The minimization procedure involves iterative calculations of $J(\boldsymbol{x})$ and its gradient $\nabla_{\mathbf{x}} J(\mathbf{x})$, using the CMAQ forward model $\mathbf{H}$ and its adjoint $\mathbf{H^T}$, as is shown in Eq. 2.

$$\nabla_x J = \mathbf{B}^{-1}(\boldsymbol{x} - \boldsymbol{x}^b) + \mathbf{H}^T(\mathbf{R}^{-1}\left[\mathbf{H}(\boldsymbol{x}) - \boldsymbol{y}\right]]) \qquad (2)$$

The algorithm that our inversion system uses to minimize the $J(\boldsymbol{x})$ is the Limited-Memory Broyden-Fletcher-Goldfarb-Shanno (L-BFGS-B) (Byrd et al., 1995), implemented in the `scipy` python module. The L-BFGS-B algorithm iteratively

adjusts $\boldsymbol{x}$ until $J(\boldsymbol{x})$ reaches a minimum. We reached a reasonable convergence for each year run after iteration 25. The ratio between the cost function and the number of observations was close to the theoretical expected value (see details Sect. 3.1). Posterior uncertainties in this study were assumed to be the same as Villalobos et al. (2020, Sect. 2.4). However, we increase their value by a factor of 1.2 to satisfy the theoretical assumption in the variational optimization (p.211, Tarantola, 1987).

## 2.2   Transport model and prior fluxes

The CMAQ model was used to simulate atmospheric transport and dispersion. These simulations, which were run off-line from the meteorological model, were conducted without atmospheric chemistry. The meteorological data used as input for the CMAQ model were taken from the Weather Research and Forecasting (WRF) model (version V4.1.1) (Skamarock et al., 2008). We run the CMAQ model at hourly resolution at a grid-cell scale of 81 km. The model has 32 vertical levels using the terrain-following $\sigma$ vertical coordinate system. Details of the parameterizations are listed in Villalobos et al. (2021, Sect. 2.4,

Table 1). We run WRF at a spatial resolution of 81 km on a single domain (i.e., non-nested). WRF initial conditions were taken from the ERA-Interim global atmospheric reanalysis (Dee et al., 2011), which has a resolution of approximately 80 km on 60 vertical levels from the surface up to 0.1 hPa. Sea surface temperatures were obtained from the National Centers for Environmental Prediction/Marine Modeling and Analysis Branch (NCEP/MMAB). The WRF model was run with a spin-up period of 12 hours.

The prior flux estimates used in our inversion consisted of four datasets: land biosphere fluxes, fossil-fuel, fires and ocean fluxes. Biosphere carbon fluxes were simulated by the Community Atmosphere-Biosphere Land Exchange model (CABLE) set-up in BIOS3 environment (hereafter referred to as CABLE BIOS3) (Haverd et al., 2018). The CABLE land surface model consists of a biophysical core, a biogeochemical module including a nitrogen and phosphorous cycle (Wang et al., 2010), the Populations-Order-Physiology (POP) module for woody demography and disturbance-mediated landscape heterogeneity

(Haverd et al., 2013b), and a module for land use and land management (POPLUC; Haverd et al., 2018). However, the functionality of POPLUC was not considered in BIOS runs, and the land-use change was held to be static at year 2000. CABLE can be run on global or regional scales. For our regional study case, CABLE was run at a regional scale (resolution 0.25 degree), and it was forced with Australian regional drivers and observations (BIOS3 set-up). Biosphere fluxes from CABLE (∼NBP) include gross primary productivity (GPP), net ecosystem respiration (autotrophic and heterotrophic respiration). However, they





do not include carbon losses from fires disturbances, harvest, erosion, and export of carbon in river flow. We used averages of 3-hourly NBP estimates as input for CMAQ (further details of how we constructed NBP can be found in Sect.2.3 Villalobos et al., 2021). The prior error covariance matrix of the terrestrial biosphere flux from CABLE was assumed to be an approximation of the net primary productivity (NPP) following the approach of Chevallier et al. (2010) with a ceiling of 3 gC $m^{-2}$ $d^{-1}$. We assumed that these uncertainties were spatially correlated with length-scale 500 km over land following Basu et al. (2013).

Within our inversion system, no temporal correlations were considered.

Fossil fuel emissions used here were based on two different inventory data sets: the Open-source Data Inventory for Anthropogenic $CO_2$ (ODIAC) (version 2019) (Oda et al., 2018) and the Emissions Database for Global Atmospheric Research (EDGAR) (Crippa et al., 2020). We added some missing sectors from the EDGAR inventory to ODIAC (such as aviation climbing and descent, aviation cruise, and aviation landing and take-off datasets). ODIAC is a global gridded product distributed at

$0.1° × 0.1°$ spatial resolution over land, which uses power plant profiles (emissions intensity and geographical location) and satellite-observed nighttime lights. We used ODIAC monthly fluxes and incorporated a diurnal scale factor to estimate diurnal $CO_2$ emission variability (Nassar et al., 2013). Given that the ODIAC product only covers the period from 2015 to 2018, we repeated the data from 2018 in 2019 but increased the value in each grid cell by 1.7%, which represents the mean annual growth rate of these emissions from 1970 to 2018. EDGAR is also gridded at $0.1° × 0.1°$ with monthly temporal resolution. There is

no EDGAR gridded product for 2016–2019, so we repeated the 2015 product to cover the other years. We increased EDGAR aviation emissions by 2.5%, which represents the mean growth rate in this emissions sector from 2016 to 2019. Fossil fuel prior uncertainties were assigned to be 0.44 times the value of the monthly fossil fuel estimates described above (see details in Sect. 2.3. in Villalobos et al., 2021). Errors in fossil fuel emissions were assumed to be uncorrelated.

Ocean flux estimates were selected from CAMS global data (version v19r1) (Chevallier, 2019). Ocean prior uncertainties

were assumed to be 0.2gC $m^{-2}$ $d^{-1}$ and uniform across the ocean, as in Chevallier et al. (2010). Similar to correlations for biosphere prior uncertainties, uncertainties of ocean fluxes were assumed to be correlated in space with length-scale 1000 km. Fire prior emissions were selected from the Global Fire Emission Database (GFED), version 4.1s, which includes emissions from small fires. Fire emissions uncertainties were assumed to be 20% of the GFED emissions and correlated in space with length-scale 500 km, but not in time. The combination of all prior fluxes was regridded to the spatial resolution of the CMAQ

model.

## 2.3   OCO-2 observations and their uncertainties

Our regional inversion assimilates satellite observations derived from NASA's Orbiting Carbon Observatory-2 (OCO-2; El-dering, 2018). The OCO-2 satellite instrument carries a single instrument that incorporates three-channel imaging grating spectrometers developed to measure reflected sunlight by the Earth's surface in three spectral bands: two $CO_2$ spectral bands

in the shortwave infrared (SWIR) at 1.6 and 2.1 µm and one in the near-infrared (NIR) ar 0.76 µm ($O_2$ A-band). From these radiance spectra is possible to calculate the column-averaged dry-air mixing ratio of carbon dioxide. OCO-2 employs three different sampling strategies to collect data: nadir, glint and target mode. Nadir observations provide useful information over land because the satellite points straight down at the surface of the Earth (surface solar zenith angle is less than 85°). In glint



mode the instrument points to the bright glint spot on Earth where solar radiation is directly reflected off the Earth's surface

(local solar zenith angle is less than 75°). In Target mode, the instrument points towards a specific location on the ground. Target mode is use for validation, where the performance of the instrument is validated against ground-based observations from the Total Carbon Column Observation Network (TCCON) (Wunch et al., 2011).

In this study, the regional inversion was performed using the combination of both land (nadir and glint) (LNLG) OCO-2 observations (version 9). We used the combination of both datasets because it has been demonstrated by Miller and Michalak

(2020) that combining both modes provides a stronger and better constraint of $CO_2$ fluxes at regional scales. Also, both datasets present negligible bias (O'Dell et al., 2018). We did not incorporate ocean glint measurements in our inversion, because ocean observations still have undetermined biases (O'Dell et al., 2018), which might impact the Australian carbon flux estimates.

OCO-2 LNLG dataset were selected from December 2014 to December 2019. We considered OCO-2 data since December 2014 because we run CMAQ with a spin up of one extra-month. Fig. A1 to Fig. A5 in Appendix A show the spatial pattern

of OCO-2 soundings (LNLG) that fall in our CMAQ domain for 2015 to 2019. In these Figures, we can see that OCO-2 data provides a very good coverage over the Australian region. Such spatial coverage offers good potential to help constrain regional biosphere $CO_2$ fluxes.

Given that the OCO-2 spatial resolution (1.29 km × 2.25 km) is higher than the CMAQ model grid cell (81 × 81 km), the OCO-2 data were averaged to the CMAQ model grid-level following a two-step process described in (Sect 2.3, Villalobos

et al., 2021). The first step involves averaging all OCO-2 soundings across 1-second intervals, while the second step involves averaging these 1-second averages into the CMAQ vertical column (approximately 11-seconds averages). The algorithm to estimate the uncertainties across 1-second averages follows Crowell et al. (2019). Here, we considered three different forms of uncertainty calculation. First, we assumed that uncertainties that fall within 1-second span were perfectly correlated in time and space (uncertainties defined as $\sigma_s$). Second, given that the average of OCO-2 uncertainties ($\sigma_s$) is relatively lower than the real

OCO-2 uncertainties (mainly because they only consider the errors from measurement noise, and not systematic errors), we also used the spread (standard deviation) of the OCO-2 retrievals in the 1-second average (uncertainties defined as $\sigma_r$). Third, we also considered a baseline uncertainty (defined as $\sigma_b$) for cases where the number of OCO-2 soundings was not enough to compute a realistic spread. Our baseline uncertainties were assumed to be 0.8 ppm over land and 0.5 ppm over ocean. Finally, we selected the maximum value between these three uncertainties ($\sigma_s$, $\sigma_r$, and $\sigma_b$). For each grid-cell, we also added

(in quadrature) to this term 0.5 ppm as the contribution of the CMAQ model uncertainty (defined as $\sigma_m$). We also increase the final observation uncertainty by a factor of 1.2 to satisfy the theoretical assumptions of the inversion (Villalobos et al., 2021). We interpolated the retrieval OCO-2 profile to the CMAQ model vertical profile as described in (Sect. 2.6., Villalobos et al., 2020). Note that we only selected OCO-2 retrievals with quality flag "0" and bias-corrected data, as described by Kiel et al. (2019).





## 2.4 Validation data

### 2.4.1 TCCON

For validation of our inversion, we compared our posterior column averaged concentration simulated by CMAQ against the TCCON sites located in Australia and New Zealand (Fig 1). TCCON is a network of ground-based Fourier Transform Spectrometers (FTS) recording direct solar spectra in the NIR/SWIR spectral region (Wunch et al., 2011). From these spectra, accurate and precise total column amounts of $CO_2$ and other trace gases are retrieved. In our study domain, there are three TCCON stations (Darwin, Wollongong and Lauder). The Darwin and Wollongong sites are located within Australia, while the Lauder site is located in New Zealand. The Darwin and Wollongong sites are operated by the Centre for Atmospheric Chemistry at the University of Wollongong, Australia (Griffith et al., 2017a, b). The Lauder site is operated by New Zealand's National Institute of Water and Atmospheric Research (NIWA) (Sherlock et al., 2017; Pollard et al., 2019). As is shown in Fig 1, the Lauder monitoring station is located on the South Island of New Zealand at 2 km north from the town of Lauder. It is sheltered from the prevailing wind direction by the Southern Alps, which increases the number of days with clear skies and results in an air mass that is largely unmodified by regional anthropogenic sources (Pollard et al., 2017). From mid-2015, the Darwin site has been located about 9 km east of Darwin city, approximately 4.5 km south-east of its previous location (Deutscher et al., 2010). The Wollongong site is a coastal site close to populated areas and industry to the north, and native forest and less dense population to the south and west (Deutscher et al., 2014). At each site, TCCON data were selected within one hour windows and averaged to be consistent with temporal resolution of the output of the CMAQ simulations. Each TCCON retrieval is provided with an averaging kernel and a prior profile, which were interpolated to the CMAQ vertical profiles. After the interpolation, we applied the averaging kernel (following Eq.15 Connor et al., 2008) to compute the TCCON CMAQ simulated $CO_2$ concentrations. The residual between CMAQ and TCCON was constructed based on monthly mean concentrations, which were calculated by taking local time averages (10:00 – 14:00 LT), where the solar radiation intensity is most stable (Kawasaki et al., 2012).

### 2.4.2 Ground-based in-situ measurements

We also compared our posterior concentrations against four ground-based in-situ monitoring sites: Cape Grim, Gunn Point, Burncluith and Ironbark, whose geographic locations are shown in Fig. 1. These monitoring sites form part of the Global Atmosphere Watch (GAW) Programme of the World Meteorological Organisation (WMO), and they are operated by CSIRO's Climate Science Centre located in Aspendale, Australia. We used hourly data from these monitoring sites, but the monthly mean averaged data shown in Section 3.5 were calculated using local time averages (12:00–05:00 LT, Australian local time, local referred to the monitoring site locations).

Atmospheric $CO_2$ concentration measurements at the Gunn Point, Ironbark and Burncluith sites are made continuously at high frequency (∼0.3 Hz) using Picarro cavity ring-down spectrometers. Instruments located at the Gunn Point and Ironbark sites use the Picarro model (G2301), while the Burncluith site uses a model G2401. All the inlets are placed at the height of 10 m. Descriptions of the Ironbark, Gunn Point and Burncluith installations can be found in Etheridge et al. (2016). Cape Grim also



operates a Picarro G2301 analyser; however, the inlet is positioned at 70 m. The instrument precision for these spectrometers
is better than 0.1 ppm (Etheridge et al., 2014) and all measurements are calibrated to the World Meteorological Organization
(WMO) X2007 $CO_2$ mole fraction scale (Zhao and Tans, 2006), ensuring comparability between all measurements used. We
note that we used "baseline" and "non-baseline" data from Cape Grim. Baseline data is selected when winds blow straight off
the Southern Ocean and have not been in recent contact with land. In this study, we used both datasets because our inversion
only uses OCO-2 soundings located over land.

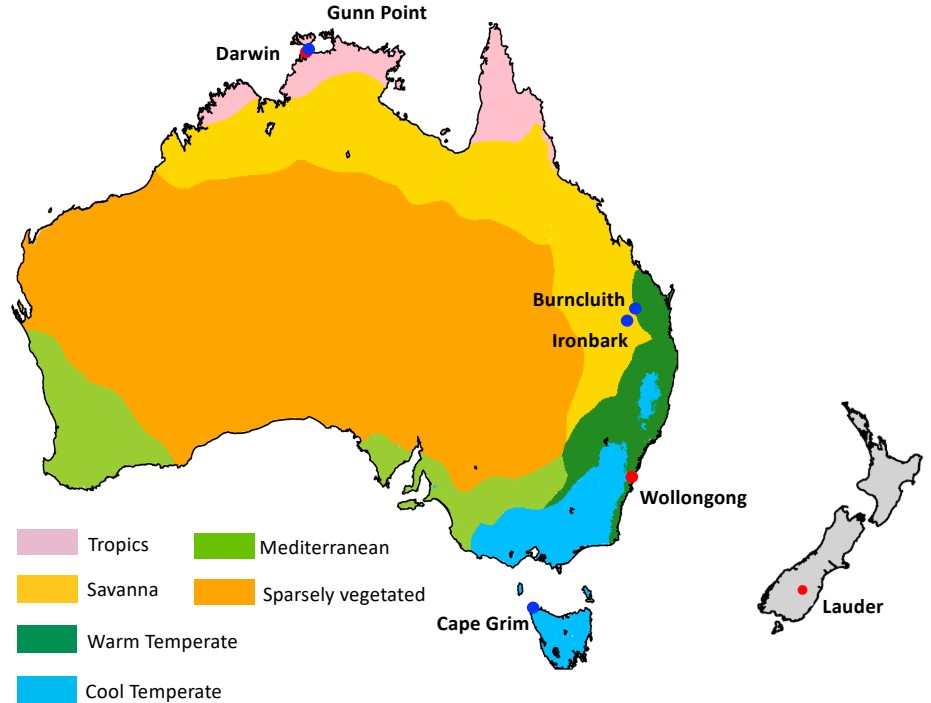

**Figure 1.** Location of the Total Carbon Column Observing Network (TCCON) sites across Australia and New Zealand (red points) and
in-situ sites (blue points). This map also shows a classification of six bioclimatic regions for Australia.

## 2.5 Australian bioclimatic classification

To understand which ecosystems contributed the most to the Australian inter-annual carbon flux variability between 2015–
2019, we divided the continent into six bioclimatic classes: tropical, savanna, warm temperate, cool temperate, Mediterranean
and sparsely vegetated (Fig. 1). We used the same six bioclimatic regions at a 0.05° spatial resolution as in Haverd et al.
(2013a). The classes were regridded over our CMAQ grid (81 × 81 km) resolution.

## 2.6 Climate data

In order to analyse the impact of climatic drivers on Australian terrestrial carbon cycle variability, we investigated the anomalies
of rainfall and temperature across Australia for the period 2015-2019. Rainfall data were selected from the Australian Water





Availability Project (AWAP), Bureau of Meteorology (BOM) (Jones et al., 2009) for the period 2015 to 2019. AWAP is a gridded product at 0.05° resolution. It is generated by spline interpolation of in situ rainfall observations. We also used air temperature data at 2 m above the land surface from ERA5, the fifth generation of European Centre for Medium-Range

Weather Forecasts (ECMWF) atmospheric reanalyses. The dataset selected from ERA5 was monthly and it was gridded at 0.25 degrees spatial resolution. We constructed 3-month running means of rainfall anomalies and air temperature anomalies relative to a mean across 2015-2019. These anomalies were calculated by subtracting their long-term mean (2015–2019) for each month from the raw time series and constructing the 3-month running mean on the resultant time series. We regridded the rainfall anomalies onto the grid of the CMAQ model in order to simplify the comparison with the estimated terrestrial carbon

uptake from the flux inversions.

## 2.7 Enhanced vegetation index (EVI) as an indicator of the vegetation greenness

Plant photosynthesis and respiration are two fundamental physiological processes in the carbon cycle. Physiological and structural changes in vegetation modulate the exchange of $CO_2$ between the land and atmosphere. In order to understand what physiological factors drive the inter-annual variability of our posterior fluxes, we studied the anomalies of the enhanced vegetation

index (EVI). EVI provides information on vegetation state, and we used it to characterize changes in Australian vegetation greenness and activity (e.g photosynthesis) from 2015 to 2019. The EVI product was derived from the Moderate-Resolution Imaging Spectroradiometer (MODIS) MOD13C1 version 6 data product, which flies on board Terra, a NASA earth-observing satellite (Didan, 2014). The MODIS EVI is a gridded product, which has a temporal resolution of 16 days composite and 0.05-degree spatial resolution. The EVI ranges from -0.2 to +1, where values less than 0 indicate a lack of green vegetation or arid

areas. We calculate the 3-month running mean of EVI anomalies in Australia relative to the long-term mean from 2015-2019 and subtract the mean seasonal cycle. These monthly EVI MODIS products were also regridded into the CMAQ domain to calculate the temporal correlation between prior and posterior flux anomalies.

## 2.8 Gross Primary Productivity (GPP)

To understand the difference between posterior and prior fluxes, we compared the climatological seasonal cycle of the gross

primary productivity (GPP) from the CABLE BIOS3 model against the remote-sensing based DIFFUSE model (Donohue et al., 2014), and the latest MODIS terra GPP product (MOD17A2H version 6) (Running et al., 2015) for the period 2015–2019. We also calculated 3-month running mean GPP anomalies for these three datasets.

The DIFFUSE GPP estimates are taken to be the product of the fraction of photosynthetically active radiation (PAR) absorbed by vegetation and the light-use efficiency. These datasets have a temporal resolution of 16-days at 250 m resolution.

Similar to the DIFFUSE estimates, the MODIS GPP product is based on a light-use efficiency approach and provides a cumulative 8-day composite product gridded at 500 m. For comparison, the CABLE BIOS3, DIFFUSE and MODIS GPP products were averaged to a monthly resolution and regridded over the CMAQ domain.





## 2.9 Global Atmospheric Inversions

We compared our Australian assimilated fluxes against nine independent global atmospheric inversions: AMES, PCTM,
CAMS, CMS-Flux, CSU, CT, OU, TM5−4DVAR, UT (Sect. 4). These global inversions are part of the OCO-2 Model Intercomparison Project (MIP) (Crowell et al., 2019; Peiro et al., 2021). In this study, we used the OCO-2 MIP flux version found in Peiro et al. (2021). In Peiro et al. (2021), the global inversions were performed using the assimilation OCO-2 (version 9, bias-corrected) from 2015-2018. A summary of these nine global inversions is given in Table 1, and a complete description of them and their input fields can be found in Peiro et al. (2021, Appendix A: model information). We can see in Table 1 that
all global inversions were run using different inverse systems and were configured at different spatial resolutions with different atmospheric transport models and prior fluxes. Some global inversion methods use a four-dimensional variational (4D-Var) approach, while others utilize the technique known as ensemble Kalman filter (EnKF) or Bayesian synthesis.

**Table 1.** Summary of the configuration of the MIP OCO-2 (version 9) design.

| Acronym | Transport Model | Meteorological fields | Grid spacing (degree) | Prior Land Biosphere | Prior Fire | Inverse System |
|---|---|---|---|---|---|---|
| AMES | GEOS-Chem | MERRA-2 | $4° \times 5°$ | CASA-GFED4.1s | GFED4.1s | 4D-Var |
| Baker | PCTM | MERRA-2 | $6.7° \times 6.7°$ | CASA-GFED3 | GFEDv3 | 4D-Var |
| CAMS | LMDz | ERA-Interim | $1.9° \times 3.75°$ | CMEMS | GFEDv4 | Variational |
| CMS-Flux | GEOS-Chem | GEOS-FP | $4° \times 5°$ | CARDAMOM | GFED4.1s | 4D-Var |
| CSU | GEOS-Chem | MERRA-2 | $1° \times 1°$ | SIB4 | GFED4 | Bayesis syntesis |
| CT | TM5 | ERA-Interim | $3° \times 2°$ / $1° \times 1°$ | CT2019 / CASA GFED4.1s | CT2019 / CASA-GFED4.1s | EnKF |
| OU | TM5 | ERA-Interim | $4° \times 6°$ | CASA-GFED3 | GFEDv3 | 4D-Var |
| TM5-4DVAR | TM5 | ERA-Interim | $2° \times 3°$ | SIB-CASA | GFEDv4 | 4D-Var |
| UT | GEOS-Chem | GEOS-FP | $4° \times 5°$ | BEPS | GFEDv5 | 4D-Var |

## 3 Results

### 3.1 Inversion performance

In our inversion system, the L-BFGS algorithm iteratively adjusts the control vector until the cost function reaches an optimal solution. In Bayesian inverse problems we require that the observational residuals (simulated − observed) and increments (posterior − background) are consistent with the assumed probability density functions (PDFs). This implies that the cost function should be approximately half the number of observations (Tarantola, 1987, p.211). Table 2 shows the analysis of





convergence for our five-year inversion. In this table, we can see that for each year, the ratio between final cost function $J(\boldsymbol{x})$

and observation was about 0.5, indicating that our system is self-consistent.

**Table 2.** Convergence diagnostics of the inversion system using OCO-2 satellite data.

| Year | $J_0(\boldsymbol{x})$ | $\nabla_x J_0$ | $J_f(\boldsymbol{x})$ | $\nabla_x J_f$ | N observations | Theoretical $J(\boldsymbol{x})$ | Ratio ($J_f(\boldsymbol{x})$ and N) |
|------|------|------|------|------|------|------|------|
| 2015 | 5653.15 | 5177.65 | 4403.01 | 394.17 | 8766 | 4383 | 0.50 |
| 2016 | 5561.51 | 5371.69 | 4380.64 | 853.07 | 8946 | 4473 | 0.49 |
| 2017 | 4485.17 | 4794.40 | 3477.63 | 335.92 | 7514 | 3757 | 0.46 |
| 2018 | 5118.29 | 3825.28 | 4112.01 | 365.46 | 9679 | 4839 | 0.42 |
| 2019 | 5582.00 | 2719.85 | 4443.32 | 387.14 | 10373 | 5186 | 0.43 |

Fig. 2 shows the monthly bias and root-mean-square error (RMSE) between the prior and posterior column integrated concentrations simulated by CMAQ against the OCO-2 observations. In this figure, we can see that the posterior concentration biases were significantly reduced from the prior to values close to zero.

In 2015, the prior concentrations significantly overestimate OCO-2 observations from March to April, and from July to

September, and underestimate the observations in November. Prior biases in these months were reduced between 60 – 90%. In March, for example, the monthly prior mean bias was reduced from 0.56 to 0.05 ppm, with a decrease in the root mean square error (RMSE) from 1.11 to 0.84 ppm. Negligible biases in the prior concentration are seen in January, February, May and December, showing a good agreement with OCO-2. In 2016, we see that the prior mainly overestimates OCO-2 observations from January to October but especially in June and August. In June, for example, we see a bias of 0.61 ppm (RMSE = 1.12),

reduced by the inversion to 0.13 ppm (RMSE = 0.78). For this year, November is the only month that shows negligible prior concentration bias, 0.04 ppm (RMSE = 1.0), indicating a good agreement with OCO-2 concentration.

In 2017, the prior concentrations simulated by CMAQ overestimate the observations from January to July. The highest prior biases are seen in April and July. For these two months, the prior biases are similar ($\sim 0.4$ ppm) with RMSE of 0.91 and 0.83 ppm, respectively. We note the data gap in August and September caused by a satellite outage. In November 2017, we see

the prior concentration underestimates the observations significantly, with biases of about -0.56 ppm and RMSE 1.29 ppm. Reduction of the prior biases in this month was about 90% (-0.06 ppm with a RMSE of 0.94).

In 2018, the prior concentration overestimates OCO-2 observations from March to May and from August to September. For this period, the highest prior concentration biases are seen in September (0.48 ppm) with a RMSE of 0.87 ppm, which were reduced by the inversion to biases of 0.17 ppm and RMSE of 0.72 ppm. From October to December, 2018, prior concentration

biases are negative.

In 2019, most of the prior biases were positive, except for January where biases were negative. For this period, prior concentration biases were -0.18 ppm, with a RMSE of 1.01 ppm, which were reduce by the inversion to -0.01 ppm and RMSE 0.87. Considerable positive biases are seen in August and September, which values were around 0.48 ppm, with a RMSE of





0.86 ppm. December was the only month that prior concentrations were in a good agreement with OCO-2 observations. Prior
biases in this period were negligible (0.08 ppm, RMSE 0.98).

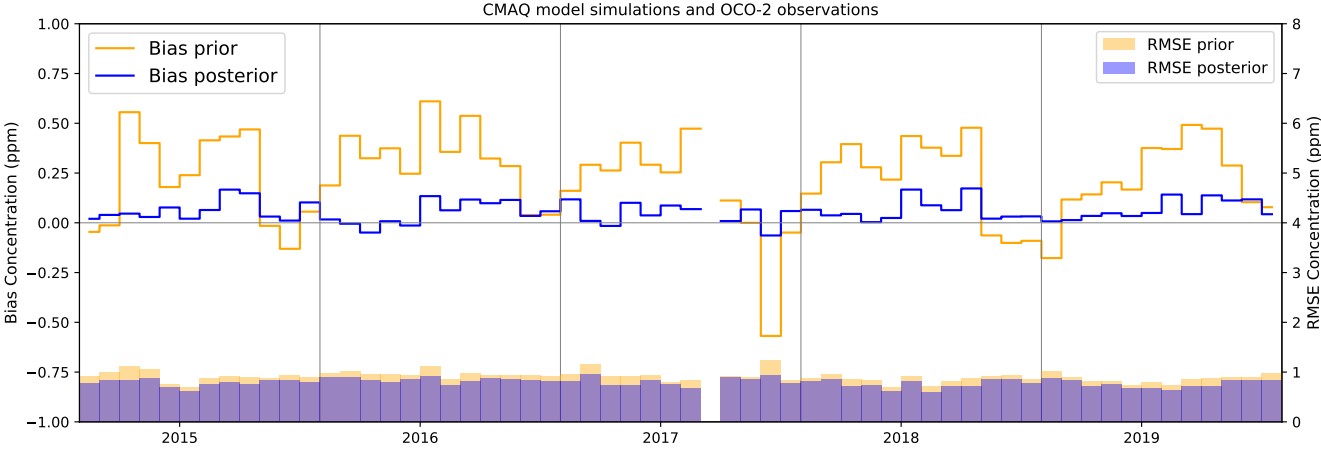

**Figure 2.** Bias and root mean square error (RMSE) between OCO-2 and the prior and posterior concentrations simulated by CMAQ model.
Blue and orange circles represent prior posterior concentration biases, and orange and blue lines represent the RMSE.

### 3.2   Australian posterior and prior monthly carbon flux estimates

Our five year inversion suggests that Australia was a carbon sink of -0.46 $\pm$ 0.09 PgC yr$^{-1}$ compared to the prior flux esti-
mate, which was 0.11 $\pm$ 0.17 PgC yr$^{-1}$. Within these estimates we only included the terrestrial part of the Australian carbon
cycle, including fires but not fossil fuel emissions. We subtracted the fossil fuel emissions from the prior and posterior estimate
because it only represents 25% of the total flux, and is subject to much less interannual variability than the biogenic fluxes.
Figure 3a shows the posterior and prior monthly mean terrestrial CO$_2$ fluxes for the period 2015–2019 along with their uncer-
tainties. Posterior flux uncertainties from 2016 to 2019 were assumed to be the same as those calculated for 2015, which were
estimated by five different observing system simulation OSSE experiments (see more details in Villalobos et al., 2021).

During the period 2015–2019, the posterior flux estimates show a stronger seasonal cycle compared to the prior flux estimate
(Fig. 3). In terms of inversion results, we see that OCO-2 sees a strong seasonal biospheric carbon uptake each year between
June and September (winter and early spring in Australia), and a stronger carbon source from November to December (late
spring and early summer in Australia). As is shown in Fig. 2, the stronger carbon uptake seen in winter and early spring occurs
because the prior concentration simulated by CMAQ model overestimate OCO-2 observations in this period.

In August 2016, we see that the carbon uptake by Australia was about -2.92 $\pm$ 0.31 PgC yr$^{-1}$ compared to the prior (-0.25
$\pm$ 0.53 PgC yr$^{-1}$). This uptake rate was the largest seen in the period 2015–2019. However, this uptake was rapidly reduced in
September (-1.09 $\pm$ 0.23 PgC yr$^{-1}$) and October (-1.27 $\pm$ 0.39 PgC yr$^{-1}$), being reversed in November (1.12 $\pm$ 0.31 PgC yr$^{-1}$)
and December (1.10 $\pm$ 0.17 PgC yr$^{-1}$). In July 2017, we again see a strong carbon uptake (-1.96 $\pm$ 0.39 PgC yr$^{-1}$), which
is rapidly reduced in September 2017 (-0.15 $\pm$ 0.23 PgC yr$^{-1}$), and eliminated in October and November 2017. In 2018, the



largest difference with the prior flux is seen from June to September. For these four months, we obtain an uptake rate of (-1.11
$\pm$ 0.3 PgC yr$^{-1}$). This carbon uptake is rapidly released back to the atmosphere in November and December. In 2019, we see
that the largest carbon uptake observed in August (-1.77 $\pm$ 0.31 PgC yr$^{-1}$) is returned to the atmosphere in December 1.50
$\pm$ 0.17 PgC yr$^{-1}$. This carbon release is the largest registered for the period 2015–2019. Later, in the discussion section, we
will show that the stronger carbon uptake is likely related to an underestimated GPP simulated by the CABLE model over the
largest ecosystem across Australia such as savanna and sparsely vegetated regions.

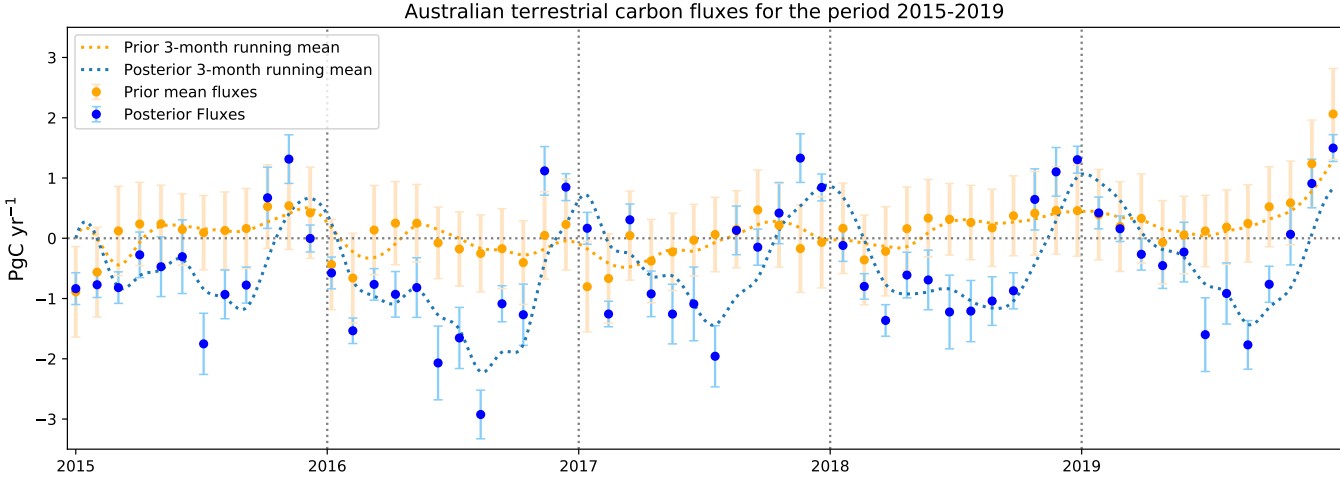

**Figure 3.** Time series of monthly mean prior (orange dots) and posterior (blue dots) carbon fluxes and their uncertainties in PgC yr$^{-1}$ over
Australia for the period 2015–2019. Uncertainties on the prior and posterior fluxes are indicated by bars.

## 3.3 Spatial distribution of posterior carbon fluxes and their anomalies

The density of OCO-2 observations allows us to localise the inferred mean uptake over 2015–2019. We plotted the spatial
patterns of the prior and posterior long-term mean flux for the period 2015 to 2019 (Fig. 4a and Fig. 4b). We can see that
the spatial pattern of the prior and posterior are broadly different. The prior mean flux distribution across Australia is almost
neutral, whereas the posterior mean flux is not uniformly distributed. The Australian south-east corner and the northern region
(with the exception of the coastal regions) are acting as sinks of $CO_2$.



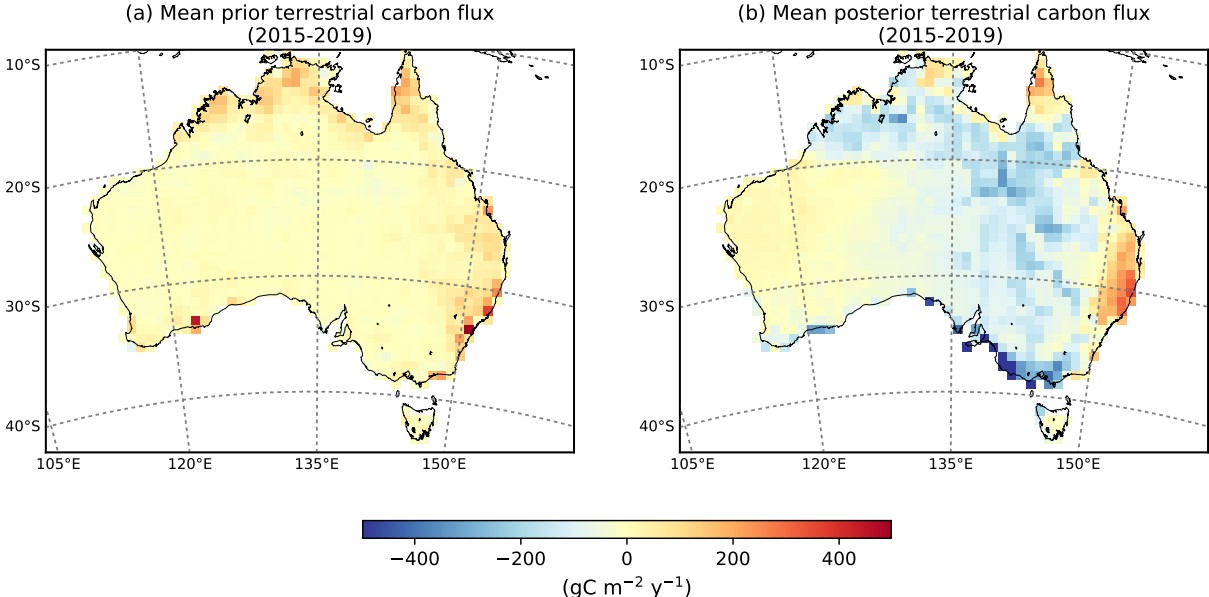

**Figure 4.** Spatial distribution map of the (a) prior and (b) posterior long-term mean terrestrial flux for 2015-2019 (gC m$^{-2}$ yr$^{-1}$) (including fires, but excluding fossil fuel emissions).

In order to understand which year contributed the most to the carbon uptake, we analysed the spatial distribution of the annual mean anomalies for 2015–2019. Fig. 5 shows the spatial distribution pattern of the annual anomalies of the posterior flux estimate in the period 2015 to 2019. A positive anomaly indicates that the amount of carbon that is released to the atmosphere is higher than the average, whereas a negative value indicates higher than average carbon uptake. This figure shows that 2015

and 2016 have noticeable negative anomalies compared to the other years. Most of the negative posterior flux anomalies seen in 2015 are distributed in the northern and south east of Australia (Fig. 5a), while the negative anomalies in 2016 are mostly distributed in the north east, central and southern region of the continent (Fig. 5b). Inter-annual variability in the terrestrial carbon cycle is mainly dependent on climate variations that impact the growing season length, regional temperatures, and soil moisture conditions, which largely depends on meteorological conditions. Despite the fact that 2016 was one of the strongest El

Niño events on record in the Pacific Ocean, the rainfall over Australia was above average for most of the continent. The annual climate report from Bureau of Meteorology for 2016 indicates that the annual rainfall over Australia was 17 per cent above the 1961–1990 average. Details of the influence of these climate driver over Australia carbon sink are explored in Section 3.3.2.

Anomalies in 2017 were mostly positive in the north-east and east region of the continent. Rainfall in 2017 was below average for much of eastern Australia, and along the west coast of Australia. The year 2019 was exceptional because it was

the hottest and driest year on record in Australia; mean temperature 1.52 °C above the 1961–1990 average average (Annual climate statement, Bureau of Meteorology, 2019). Positive posterior flux anomalies occur all over the continent, except on the south-east and north-east corner of the region. It is likely that the higher carbon release to the atmosphere in this period was



largely affected by the drought that covered large parts of the country in 2019. These conditions were also intensified by the wildfires, which were the worst on record.

**Figure 5.** Spatial distribution maps of the annual posterior terrestrial $CO_2$ flux anomalies (gC m$^{-2}$ yr$^{-1}$) for 2015 to 2019 (panels a, b, c, d and e). Negatives anomalies correspond to a larger than average uptake of $CO_2$ by land ecosystem, whereas positives anomalies correspond to a larger than average release of $CO_2$ to the atmosphere from the land.





### 3.3.1 Australia's vegetation greenness anomalies and land carbon flux anomalies

Fig. 6 shows the time series of the 3-month running mean of prior and posterior flux anomalies against 3-month running mean of EVI anomalies for the period 2015 to 2019 classified by six agro-climate ecosystems: tropics, savanna, warm temperate, cool temperate, Mediterranean and sparsely vegetated (Fig. 1). In general for each category, both the prior and posterior flux anomalies were negatively correlated with EVI anomalies (Fig. 6d), which indicate that an increase in vegetation productivity is associated with an increase of the carbon uptake by the terrestrial ecosystem.

It is evident that over the tropical ecosystem, there is not much variability of the posterior and prior fluxes throughout 2015–2019 (Fig. 6a). In this category, we only see significant posterior negative anomalies in December 2017 (-0.21 PgC yr$^{-1}$) and January 2018 (-0.11 PgC yr$^{-1}$). This larger than average uptake of carbon coincides with significant positive EVI anomalies (>0.04) in this period. We also see significant positive EVI anomalies from November 2018 to February 2019. However, for this whole period, the positive EVI anomalies only coincide with the posterior anomaly in February 2019. In this category, the temporal correlations between posterior flux anomalies and EVI anomalies were weak (R= -0.23) compared to the prior flux anomalies (R = -0.38).

Over the savanna ecosystem, our posterior flux anomalies show considerable variability throughout 2015–2019 (Fig. 6b). In 2015, from May to August, we see that our positive posterior flux anomalies coincide with negative EVI anomalies in that period. These anomalies disappear from October to November and turn negative in December 2015 (-0.4 PgC yr$^{-1}$). The anomalies seen in December extend until February 2016. This higher than average carbon uptake, coincides with positive EVI anomalies in that period. These posterior flux anomalies again vanish in March 2016, as a consequence of lower than average vegetation productivity (negative EVI anomalies). Positive EVI anomalies are again seen from April to June 2016, which coincide with negative posterior carbon anomaly in this period ($\sim$ -0.35 PgC yr$^{-1}$). In 2017, we see that the positive EVI anomalies recorded from January to April do not align with the larger than average posterior carbon release in this period. We believe that the few OCO-2 soundings found in this period limit the potential of our inversion to constraint the surface fluxes in the savanna ecosystem (see Appendix A, Fig A3, panels a, b and c). We found similar results in September and November 2017. As mentioned in Section 3.1, there was a long data outage of 51 days from August to September 2017. In September the number of OCO-2 observations in Australia were only 221 and most of soundings were seen over sparsely vegetated ecosystem (in central Australia) than the savanna ecosystem (see Appendix A, Fig A3, panels h, i). In this category, the temporal correlation between the posterior anomalies and EVI anomalies was moderate (R= -0.38) compared to the prior flux anomalies and EVI anomalies (R = -0.52). We note that the time series temporal correlation between EVI anomalies and posterior anomalies are better represented at grid-cell scale resolution (Fig. 9a) than the time series correlations calculated within the ecosystem because these are not influenced by high spatial variability within the ecoregion. We can see in Fig. 9a that temporal correlations at grid-cell scale resolution in the north-east region of the savanna ecosystem reach values of about 0.75.

The warm temperate ecosystem does not show as much variability as savanna (Fig. 6c). In this ecosystem, we only see significant negative posterior anomalies in 2016 and 2017. For 2016, the posterior anomalies spanned August–October, with





average value -0.2 PgC yr$^{-1}$. This coincides with positive EVI anomalies registered in that period (0.06). These anomalies
also exist in the prior but the inversion has brought them forward in time. During 2017 we also observe negative posterior
flux anomalies but not as strong as 2016. Negative posterior anomalies span March–June, with value -0.1 PgC yr$^{-1}$. These
also coincide with positive EVI anomalies (0.04). For 2018, from June to October, we mostly see positive posterior anomalies,
which appear to be associated with the negative EVI anomalies (-0.05). No posterior variability was seen in 2019 with the
inversion reducing anomalies in the prior. Temporal correlation between the posterior anomalies and EVI anomalies in this
category were moderate (R = -0.34).

As mentioned at the beginning of this section, we did not find correlations between posterior anomalies and EVI anomalies
over the cool temperate category. But, we did find a moderate correlation between prior anomalies and EVI anomalies (R
=-0.36) (Fig. 6g). We believe that this discrepancy is due to difficulty in estimating the posterior fluxes along coastal areas.
We should note that this category is extended in the south-eastern corner of Australia, which is strongly affected by strong
winds that come from the southern ocean, and cloudy conditions that prevent to retrieve satellite observations in this area.
For the majority of the months between 2016 and 2017, the number of OCO-2 sounding in this category was relatively low
compared with the other years (see Appendix A; Figs A2 and A3). For example, the reduced number of soundings found in
period (May–December) in 2016 might explain why despite having positive EVI anomalies, we did not see a carbon uptake in
this period.

Posterior variability over the Mediterranean ecosystem is small throughout 2015 and 2019 (Fig. 6f). We only see some
variability in the posterior anomalies in 2016 and 2017. In 2016, from January to June, we note only small negative posterior
anomalies of about -0.09, which coincide with the positive EVI anomalies in that period (0.06). Note that the relative small
variability in this ecosystem is because we did not standardize the fluxes per unit area. Results recorded in 2017 show that
despite having a higher than average EVI in the period January to April, we observe a release of carbon of about 0.1 PgC yr$^{-1}$
in this period. Looking at Fig. 5, we observe that most of the positive posterior anomalies come from coastal areas in the
Mediterranean ecosystem, and prevailing winds at the coastal zone restrict the ability of OCO-2 to constrain surface fluxes
(Villalobos et al., 2020). The years 2018 and 2019 were mostly impacted by negative EVI anomalies but only small posterior
flux anomalies. Over this category, the temporal correlation between posterior flux anomalies and EVI anomalies was weak
(R= -0.21) compared to the correlation between prior flux anomalies and EVI anomalies (R = -0.33).

Interesting findings are seen in 2016 and 2017 over sparsely vegetated areas. From August 2016 to April 2017, we found this
biome contributed enormously to the long-term mean sink estimated between 2015 and 2019. After April 2017, the significant
carbon sink anomaly disappears, and it turns into a positive carbon flux anomaly. These findings were also consistent with a
decrease of the greenness of the vegetation. In 2018 and 2019, we only see positive carbon flux anomalies, which again align
with negative EVI anomalies.





**Figure 6.** Time series of 3-month running mean posterior (black line) and prior (grey line) terrestrial $CO_2$ flux anomalies (PgC yr$^{-1}$) and 3-month running mean EVI anomalies (turquoise dashed line) between 2015 to 2019 aggregated by agro-climate regions (a) Tropics, (b) Savanna, (c) Warm temperate, (d) Cool temperate, (e) Mediterranean and (f) Sparsely vegetated ecosystem. The grey shaded area represent 1.0 standard deviation range around the mean for the prior and posterior flux uncertainty.

### 3.3.2 Climate drivers of Australian land carbon flux anomalies

We saw in the previous section that the variability of the prior and posterior flux anomalies is associated with the variability of vegetation greenness (EVI as proxy of the vegetation productivity). A higher land productivity (i.e., positive EVI anomalies) leads to a higher uptake of $CO_2$, and negative EVI anomalies cause the opposite effect. Such variability is more evident in some ecosystems than others. We observed that savanna and sparsely vegetated ecosystems account for the highest variability of our posterior fluxes. We also see some variability of the posterior fluxes over warm temperate, but not as strong as the categories





mentioned above. In this section, we will examine how changes in rainfall and temperature drive most of the variability in these ecosystems, with a focus over the savanna, warm temperate and sparsely vegetated regions.

Fig 7 shows 3-month running mean of the prior, posterior flux anomalies and 3-month running means of rainfall anomalies for the period 2015 to 2019 classified by the same six ecoregions discussed in section 3.3.1. Fig. 8 shows the flux anomalies
along with 3-month running mean of temperature anomalies. In general, we found that rainfall and temperature anomalies are negatively correlated, and Australia's semi-arid ecosystems are highly responsive to these climate drivers. It is evident that above-average rainfall and low temperatures lead to unexpectedly rapid vegetation growth over semi-arid ecosystems shown in Fig. 6 making a significant contribution to the removal of carbon dioxide from the atmosphere.

We see in Fig. 7b that the savanna ecosystem was generally influenced by negative rainfall anomalies in 2015, which
coincides with the positive posterior flux anomalies in that year. December was the only exception in 2015, because the amount of rainfall in this month was above average (25 mm), which aligns with the timing of the wet season in northern Australia. An increase in the amount of rainfall might explain the larger than average negative posterior anomalies in this period. We see that the posterior negative flux anomalies remain in January 2016, but its intensity decreases as the amount of rainfall also decrease. Negative rainfall anomalies in 2016 are only seen from February to April. In the period May 2016 to December
2017, we again notice an increase of the amount of rainfall, which coincides with the anomalous posterior sink in that period. For this category, the temporal correlation between the posterior flux anomalies and rainfall anomalies was weak (R = -0.15). However, we plotted the temporal correlation calculated at each grid-point, and we discovered that in some areas over savanna, correlations are strongly negative, with values of about -0.4 to -0.8 (Fig. 9b).

In regards to temperature anomalies over savanna, we can see in Fig. 8b that they were below average by -0.5°C in 2015.
Despite having estimated negative temperature anomalies recorded in 2015, we do not see a carbon sink anomaly in this ecosystem as we expected. We believe that the savanna ecosystem responded strongly to the deficit of water seen in this period compared to the relatively small changes in the air temperature. Positive temperature anomalies recorded from March to June in 2016 (2°C above average) lead to an increase in the amount of carbon released to the atmosphere in that period. Temperature in 2017 was predominantly below average from January to June. In 2018, from February to March, we mostly
found temperatures that were below average by -1°C. In this period, negative temperature anomalies in combination with positive rainfall anomalies align with higher posterior sink recorded in that period. Air temperature anomalies in 2019 were only positive. The highest temperature anomaly recorded was in June and November with values of about 2°C, which coincided with the positive posterior flux anomalies in this period.

The warm temperate ecosystem was also affected by higher rainfall anomalies which were more significant in years 2016 and
2017. Positive rainfall anomalies (50 mm above average) might explain the stronger carbon uptake estimated by the inversion from July to December, 2016 and from May to March, 2017. In 2018 to 2019 we only see negative rainfall anomalies in this category, which again line up with a higher than average carbon released by the ecosystem. Temperature anomalies in this category were mostly negative from 2015 to 2018. The air temperature in 2019 was generally above average in this category. Positive air temperature anomalies (4 °C above average) coincide with the higher than average release of carbon recorded by
the ecosystem in this period.





We also see climate relationships in 2016 and 2017 over sparsely vegetated areas (Fig. 7f). We see that the amount of rainfall in this category is above average from August 2016 to April 2017. These positive rainfall anomalies (25-30 mm) coincides with the negative posterior flux anomalies estimated in that period and with negative air temperature anomalies of about -1°C to -2°C. From May 2017 on-wards the positive precipitation anomalies are cancelled out and shift to being negative until the end of 2019. This deficit of rainfall coincides with a change in air temperature anomalies in this period.

In 2018, no significant changes in temperature were observed. In 2019, positive temperature anomalies led us to expect an increase in the ecosystem respiration. In this ecoregion, the time series temporal correlation between posterior anomalies and temperature anomalies was positive (R = 0.34). However, if we look at temporal correlations calculated at each grid-point in Fig. 9, they are stronger (R = 0.5–0.9). Spatial averaging smooths grid point anomalies and so dilutes signals.

**Figure 7.** Time series of 3-month running means for anomalies of rainfall (light-blue dashed line) and terrestrial $CO_2$ fluxes between 2015 to 2019. Details of the other components of the Figure are the same as shown in Fig. 6.





**Figure 8.** Time series of 3-month running means for anomalies of temperature (orange dashed line) and terrestrial $CO_2$ fluxes between 2015 to 2019. Details of the other components of the Figure are the same as shown in Fig. 6.







**Figure 9.** Spatial map of monthly temporal correlation between (a) EVI and posterior anomalies (b) rainfall anomalies and posterior anomalies, and (c) air temperature anomalies and posterior anomalies for the period 2015–2019.



### 3.4 Comparison with TCCON measurements

Figs. 10 and 11 show the comparison between the monthly mean column-averaged from TCCON sites and the prior and posterior column averaged concentration simulated by CMAQ. Monthly averages were computed using data selected between 10:00− 14:00 (Australia and New Zealand local time).

We can see in Fig. 10a that the posterior column-averaged $CO_2$ simulated by CMAQ and the observations at the TCCON Darwin site shows seasonal differences for some periods from 2015 to 2019. Assimilating OCO-2 data only slightly reduced prior bias. An exception is in the spring and summer seasons of 2015 and 2016. For example, in November and December 2015 we found a reduction of the biases of about 20 to 90%. We could not compare these results with November and December in 2016 and 2018 because this TCCON site did not record data in this period. In November 2017 and 2019 the percentage of reduction of prior biases were not as significant as 2015 (26 and 16%). High negative posterior biases are seen in February and March in 2018 and 2019. In February 2018, for example, we found biases of about -1.83 (RMSE = 1.95), which are similar to the biases found in the same period of 2019. These negative biases may be related to the small number of OCO-2 soundings located around the site (see Appendix A, Figs A4 and A5) or local biases in the OCO-2 data (Peiro et al., 2021). The small number of OCO-2 observations around Darwin is due to the presence of cloud cover and aerosols. While in summer, Northern Australia experiences a wet season (November to April), which is highly impacted by monsoonal rains and storms. Winter, the dry season in this region, is affected by fires. Some studies (i.e. Taylor et al., 2016) suggest that some OCO-2 retrievals can be biased by clouds during the wet season and smoke aerosol plumes during the dry season, mainly because the OCO-2 cloud screening algorithms present some difficulty in identifying clouds near the surface. With regard to correlation analysis, we also found that the relationship between observations and the posterior simulations is improved in some periods (see Appendix A, Table B1).

Evaluation at the Wollongong site (Fig. 10b) also shows systematic differences with our posterior concentrations. From 2016 onwards, we see a persistent underestimation of the prior and posterior column average simulated by CMAQ. Similar to Darwin, the posterior estimates derived from the inversion does not help much to reduce the prior biases at this site. In general, we see the prior and posterior remains almost the same (biases are less than 1 ppm), except in winter 2015, where biases are about 1.5 ppm. Considerable reduction of the prior biases are only seen in summer 2016/2017 (November and December), where the prior biases were reduce by 20 and 80%. As discussed in Villalobos et al. (2021), the improvement in bias is negligible when the wind blows from the ocean to this site or not many OCO-2 soundings were found around the monitoring location. Improvement of correlation at Wollongong site are shown in Appendix A, (Table B2).

Unlike Wollongong site, we see a persistent overestimation of both the prior and posterior estimates at TCCON Lauder (Fig. 10c). Posterior biases are, however, less than 1 ppm. Prior biases at the Lauder site were mainly reduced in winter and early spring. The reduction of the biases at this site were modest (about 10 to 25%). New Zealand is (relative to the Australian mainland) much smaller and narrower along the south-west to north-east direction, and thus strongly affected by oceanic airflow. The smaller size means relatively few OCO-2 soundings are retrieved over this area. Ocean fluxes that affect




New Zealand have less freedom to be modified due to the small prior uncertainties assumed in the inversion. Analysis of the correlation is shown in Appendix A, Table B3.

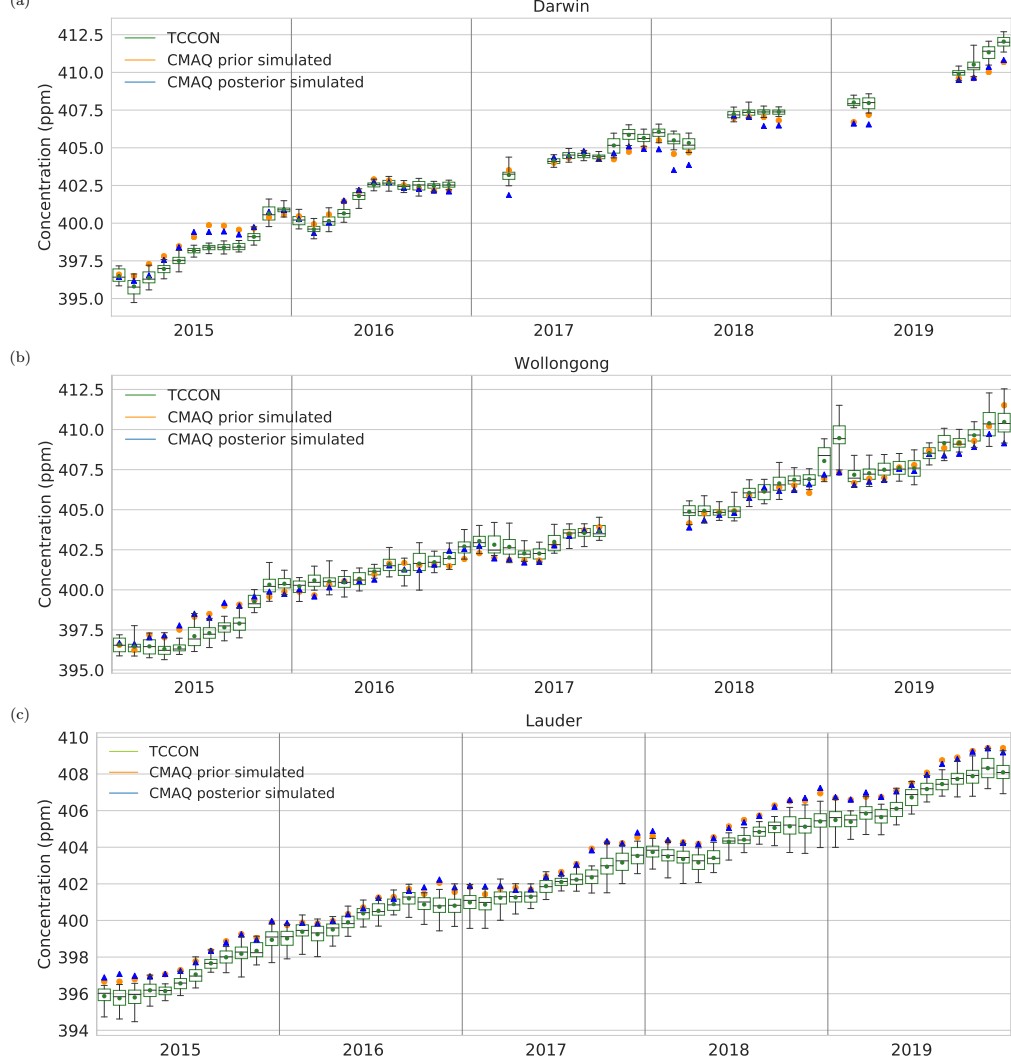

**Figure 10.** Comparison between the monthly mean column-averaged at (a) Darwin, (b) Lauder, and (c) Wollongong TCCON sites and the CMAQ prior and posterior modeled $CO_2$ concentrations for 2015–2019. Box plot diagrams represent TCCON sites (green dots indicates the mean and the horizontal green line the media). The top edge of the box represents the 75th percentile, and the bottom edge represents the 25th percentile. The top and bottom whiskers represent the 95th and the 5th percentile. Orange dots represent the prior mean concentration, while the blue triangle represents the posterior concentrations.





**Figure 11.** Bias and root mean square error (RMSE) between TCCON and the prior and posterior modeled concentrations at TCCON sites (a) Darwin, (b) Wollongong and (c) Lauder for 2015–2019. Blue and orange circles represent prior posterior concentration biases, and orange and blue bar represent the RMSE.




### 3.5 Comparison with ground-based in situ measurements

Figs. 12 and 13 show the comparison between ground-based in situ measurements (Gunn point, Burncluith, Ironbark and Cape Grim) and the prior and posterior simulated by the CMAQ model at the surface for 2015–2019. Averages were computed using data selected between 12:00 − 17:00 (Australia local time).

We note in Fig. 12a that there is a gap in the data at Gunn Point from June 2016 to August 2018. Our comparison with the rest of the months shows that our posterior simulations do not represent the Gunn Point site very well. In general, our posterior concentrations are underestimating the observations. The prior concentration indicates a better agreement, but biases are still significant. Some possible explanations for these results might be related to the limited vertical resolution of these retrievals and consequently the relative inability of OCO-2 to constrain fluxes at the scale relevant to this site (total column measurements are less sensitive at the surface than in-situ sampling). Another possible explanation is that within our model, Gunn Point is a coastal site, which is affected by prevailing offshore winds. If winds come from the ocean our fluxes are less constrained by OCO-2 retrievals (see plot of wind directions Supplementary; Figs. S17 and S20 January −February).

The data for the Burncluith site spans July 2015 to May 2018 (Figs. 12b, 13b). In 2015, we see that our posterior concentrations at the surface do not represent very well the Burncluith site. However, in 2016 and 2017, we do see some improvement of the biases in some periods. For example, we note that prior biases are reduced in summer (e.g., January and February) and spring (e.g., September and October). In these seasons, improvements of the biases are greater than 50%. No improvement of the biases are seen in June, July and August (winter season in Australia) 2016. In this period, the prior concentrations show a better agreement with the observations, with biases ranges between 0.54 to -0.46 ppm compared to the posterior biases range between -2.79 to -3.57 ppm). One possible explanation for these results could be associated with errors in the transport of the CMAQ model, such as a misrepresentation of the wind flows or errors related to the vertical mixing scheme. Transport errors related to vertical mixing are often associated with atmospheric turbulence near the surface where most $CO_2$ observations are made. Therefore, correcting the prior column average simulated by CMAQ to match OCO-2, the correction imposed by the inversion near the surface might be wrong. For 2018, prior biases were not reduced but they were small (range -0.16 to 0.46 ppm), except February (-2.31 ppm) compared to posterior biases. Again, we suggest that high posterior biases might be related to the vertical transport of the CMAQ model. We also did not find much improvement in the correlations at this site (see Appendix A, Table C4).

Similar to Burncluith, Ironbark also has gaps in the data (Fig. 12c). The data for the Ironbark site spans January 2015 to November 2016. Subsequently there was only data for January 2017 and February and March 2018. Results for the Irobark monitoring station are similar to Burncluith. These results were not unexpected given the stations' proximity (Fig. 1). For 2015 and 2016, validation against Ironbark site also shows that the posterior mean concentrations were in good agreement with the observations for summer and spring season. In November 2015, for example, the prior mean bias was reduced from -2.28 ppm (RMSE = 2.86) to -0.05 ppm (RMSE = 1.93). In Winter 2015, the posterior biases got worse. In July 2015, for example, prior biases were -0.35 ppm (RMSE = 1.66) and the posterior biases were -2.33 ppm (RMSE = 2.77). In February 2018, we see the posterior biases were about -11.4 (RMSE = 21.61). The difference between the posterior simulation and the observations





may be related to a single event visible to the surface station but not seen by OCO-2. On February 7th, Ironbark registered a

535     concentration of $CO_2$ of 459.87 ppm. It is possible that fires may have caused the high $CO_2$ concentration registered in this

period (see information for February, 2018 at NASA Fire Information for Resource Management System (FIRM, 2020). From

April 18th to 20th , we also see a similar event that was not captured by the inversion, causing a posterior concentration bias of

-6.44 ppm (RMSE = 9.82). During these 3 days the concentrations registered at Burnlcluith was greater than 450 ppm.

Cape Grim is the only site with a complete time-series of observations during this period (Fig. 12d). Like Gunn Point, Cape

540     Grim is a coastal site, and it is affected by strong westerly winds that blow from the ocean into Tasmania. We can see in Fig. 12d

that there is an evident underestimation of our posterior fluxes across 2015 to 2019. However, there are some months in 2015,

2016 and 2017 that we see a significant reduction in the prior bias. In May 2015, for example, reduction of the biases were

87%. For November 2016 to April, 2017 the reduction of the biases is more noticeable. In April 2017, for example, we found

a reduction of the biases of about 70%. Stable winds in the period might be associated with the improvement of the biases (see

545     Supplementary; Fig. S33). In general, all the negative large posterior biases for all the months ($2-5$ppm approximately) are

associated with the the strong westerly and north-westerly winds that come from the ocean to Tasmania. As mentioned before,

Cape Grim is a coastal station, whose aim is to record clean air that blows from the southern ocean and it is not representative

of Tasmania's air mass.






**Figure 12.** Comparison between monthly mean $CO_2$ concentrations (ground-based stations) at (a) Gunn point, (b) Ironbark, (c) Burncluith, and (d) Cape Grim sites and the CMAQ prior and posterior modeled $CO_2$ concentrations for 2015–2019. For details of what the different components of the box-plot represent, see the caption of Fig. 10.





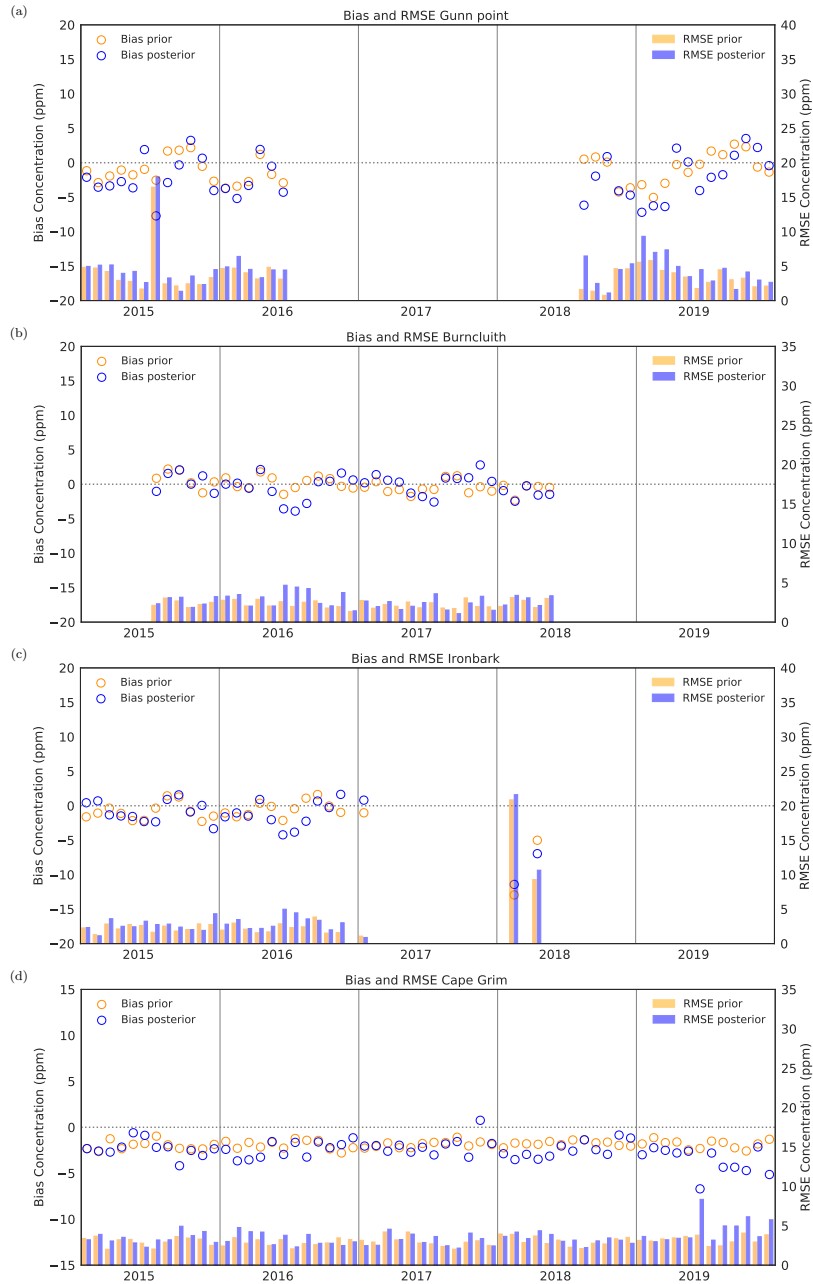

**Figure 13.** Bias and root mean square error (RMSE) between surface measurements and the prior and posterior modeled concentrations at (a) Gunn point, (b) Burncluith, (c) Ironbark and (d) Cape grim for 2015–2019. Blue and orange circles represent prior posterior concentration biases, and orange and blue bar represent the RMSE

.



## 4 Discussion

**Comparison between the seasonal cycle of the prior, posterior, and GPP fluxes**

To further examine the difference between the prior and posterior terrestrial fluxes shown in Fig. 3, we calculated the climatological seasonal cycle of the prior, posterior (Fig. 14) and the gross primary productivity (GPP) fluxes derived from CABLE BIOS3, MODIS and the DIFFUSE model (Fig. 15) for the period 2015 and 2019 aggregated by the same six bio-climate ecosystems described in Section 3.3.1.

It is evident that savanna (Fig. 14b) and sparsely vegetated (Fig. 14d) ecosystems are the regions that show the largest difference between prior and posterior flux estimates. Over the savanna region, the most evident difference is seen from June to September. The absolute difference between prior and posterior fluxes in this period is about 0.4 to 0.5 $PgC\ yr^{-1}$. According to MODIS and DIFFUSE GPP estimates, the stronger posterior sink observed in this period may be due to an underestimation of GPP simulated by the CABLE BIOS3 model (Fig. 15). The GPP estimated by MODIS from June to September was about 0.90 $PgC\ yr^{-1}$ compared to CABLE BIOS3, which was 0.59 $PgC\ yr^{-1}$. DIFFUSE GPP estimates were around 0.68 $PgC\ yr^{-1}$.

Over the sparsely vegetated region, the seasonal discrepancy between the prior and posterior flux is more evident than for savanna. The seasonality of the posterior flux is stronger (relative to the prior estimate) from April to September. In this ecosystem, the largest absolute difference between the prior and posterior fluxes is seen from June to August (0.5 $PgC\ yr^{-1}$). In July, for example, the inversion shifts the prior flux from -0.05 $\pm$ 0.09 $PgC\ yr^{-1}$ to -0.56 $\pm$ 0.06 $PgC\ yr^{-1}$. Again, we can see in Fig. 14d that a possible reason of this shift may be associated with an underestimation of the GPP by CABLE BIOS3. It is evident that MODIS and DIFFUSE GPP have a stronger seasonality compare to CABLE BIOS3 GPP. For example, from June to August, the CABLE BIOS-3 GPP was about 0.4 $PgC\ yr^{-1}$ compared to DIFFUSE and MODIS, which were 0.9 and 1.3 $PgC\ yr^{-1}$ respectively. We did not find a seasonal correlation between the prior fluxes and MODIS and DIFFUSE GPP fluxes (see Appendix D, Table D1), but we did find a positive correlation between the posterior fluxes and the GPP estimated by MODIS and DIFFUSE (R = 0.44 and R = 0.45 respectively).

Regarding the Tropics, warm temperate, cool temperate, and Mediterranean ecosystems, the seasonal correlation MODIS or DIFFUSE GPP estimates were stronger for the prior than for the posterior fluxes (Appendix D, Table D1). A stronger correlation between the prior flux and MODIS and DIFFUSE might be attributable to the fact that the assimilated coastal fluxes might be somehow less constraint by the inversion in these ecosystems, mainly because they are mostly influenced by ocean fluxes where the uncertainties have less freedom to be modified by the inversion.







**Figure 14.** Climatological seasonal cycle of prior (orange points) and posterior (blue points) terrestrial fluxes (2015–2019).



**Figure 15.** Climatological seasonal cycle of GPP (2015–2019) derived from CABLE BIOS3 model (yellow-green dashed line), MODIS (forest-green dashed line), and DIFFUSE model (lightcoral dashed line).





**Comparison between CABLE BIOS3 GPP and the GPP anomalies derived from MODIS and DIFFUSE model**

We saw in the previous section that over the sparsely vegetated and savanna ecosystems, the seasonal cycle of our posterior flux estimate shows a larger amplitude compared to the prior estimate. We suggested that the amplitude difference between these fluxes may be due to an underestimate of GPP by the CABLE BIOS-3 model. In this section, we investigate whether

the CABLE BIOS3 model is also underestimating the GPP flux anomalies. We did this analysis because even though the prior and posterior flux anomalies, in general, follow similar patterns in each bioclimatic ecosystem, the amplitudes of these two flux anomalies estimates show discrepancies. Fig. 16 shows the time series of GPP flux anomalies for the period 2015 to 2019 classified by the same six bio-climate ecosystems described in Section 3.3.1 against the 3-month running mean of prior and posterior flux anomalies.

Over the savanna ecosystem (Fig. 16b), we see that the negative posterior flux anomaly shows a larger amplitude than the prior anomalies. We found that the stronger posterior negative flux anomaly recorded from August to November 2016 is not due to CABLE BIOS3 GPP flux anomaly underestimation. On the contrary, from August to November 2016, we see that CABLE BIOS3 GPP (positive anomalies) overestimate MODIS and DIFFUSE GPP. MODIS and DIFFUSE GPP values were similar (0.21 and 0.22 PgC yr$^{-1}$ respectively) compared to the CABLE BIOS3 GPP (0.30 PgC yr$^{-1}$). In the opposite direction,

we observed negative GPP anomalies in the summer period (January to April) in 2019. We see that the CABLE BIOS3 GPP flux anomalies are even more negative than MODIS and DIFFUSE, showing a lower greenness in vegetation than the average over this ecosystem. In March 2019, for example, the CABLE BIOS GPP anomalies were -0.56 compared to MODIS and DIFFUSE, which were -0.32 and -0.27, respectively. An unanswered question in this study is why our posterior flux anomalies show a larger carbon release from this ecosystem compared to the prior anomaly if CABLE BIOS3 GPP flux anomaly is more

negative than MODIS and DIFFUSE GPP anomalies? Could it be possible that due to a deficit of water and high temperatures recorded in 2019, the respiration from soils and vegetation is being overestimated by the CABLE model, causing low GPP in the savanna ecoregion? Further analysis of the terrestrial ecosystem respiration is needed to understand the variability and trend of the sources and sinks of this ecosystem. In this ecosystem, the correlation between prior flux anomaly and the CABLE BIOS3 GPP is -0.73 compared to the correlation of posterior anomalies and CABLE GPP, which was -0.50.

Over the sparsely vegetated ecosystem (Fig. 16f), we found that the GPP anomalies from CABLE BIOS3, MODIS and DIFFUSE show similar patterns. However, we found that the correlation between the posterior anomalies and CABLE BIOS3, DIFFUSE and MODIS GPP anomalies were stronger than the one correlated with prior. For example, the correlations between the prior and CABLE BIOS3 GPP anomalies were -0.46 compared to the posterior (R = -0.61). Correlations between the posterior anomalies and DIFFUSE and MODIS GPP was also stronger than the prior, which value was -0.5 compared to the

prior (R = -0.3) (more details in Appendix D, Table D2). These findings are significant for Australia because they suggest that our OCO-2 inversion might likely be better at capturing the anomalies of this ecosystem (the largest ecosystem in Australia) compared to the biosphere land model ecosystem.

Overall, we found that the savanna and sparsely vegetated ecosystems dominate the inter-annual variability for the period 2015–2019. These results are consistent with Poulter et al. (2014) and Ahlström et al. (2017), who found that semi-arid



ecosystems such as Australia are highly responsive to climate drivers such as rainfall and temperature. The result supports the broader view of carbon-cycle/climate interactions than the original focus on the wet tropics (e.g. Rayner and Law, 1999; Bousquet et al., 2000).

**Figure 16.** Time series of 3-running monthly mean GPP anomalies derived from CABLE BIOS3 model (yellow-green dashed line), MODIS (forest-green dashed line), and DIFFUSE model (orange dashed line) between 2015 and 2019.



### Comparison with the OCO-2 MIP global inversion

In Sect 3.1, we saw that validating the posterior concentrations against the current Australian GHG monitoring system is

challenging, mainly because Australia has a small number of monitoring sites, especially given its large landmass. In addition, the locations of these sites are far away from the bioclimatic regions, where the OCO-2 inversion suggests a larger carbon sink relative the prior flux, such as the savanna and regions with sparse vegetation.

We compared our results against the monthly carbon flux mean of nine independent global OCO-2 (LNLG) global inversions (AMES, PCTM, CAMS, CMS-Flux, CSU, CT, OU, TM5−4DVAR, UT) for 2015-2018 (see Fig. 17). The annual OCO-2

ensemble mean of carbon fluxes shown in Fig. 17 suggest that Australia was a carbon sink of -0.26 $\pm$ 0.22 for 2015–2018, similar to our posterior flux estimate (-0.52 $\pm$ 0.08 PgC yr$^{-1}$). The MIP uncertainty is calculated from the spread of the ensemble mean of the nine models. In terms of seasonality, we can observe in Fig. 17 that for several periods between 2015 and 2018, the monthly mean of our posterior carbon fluxes falls within the uncertainties of the OCO-2 MIP ensemble mean, except for some months in winter. For example, we notice that the large carbon sink estimated by our inversion (-2.92 $\pm$ 0.27

PgC yr$^{-1}$) in August 2016 does not fall within the ensemble monthly MIP mean of that period (-1.28 $\pm$ 0.78 PgC yr$^{-1}$). However, the carbon flux estimate derived by PCTM (-2.31 PgC yr$^{-1}$) and CSU (-2.66 PgC yr$^{-1}$) global models shows similar results to our flux estimate. These findings are also found throughout 2015, 2017 and 2018, where our posterior carbon flux estimates closely follow PCTM and CSU seasonal patterns.

We also studied the carbon flux anomalies derived by the OCO-2 MIP and compared them with the prior and posterior flux

anomalies (3-month running mean) that we have discussed throughout this study (Fig. 18). We see in Fig. 18 that both prior and posterior carbon flux anomalies have similar variability to the majority of the models in MIP. Furthermore, it is evident that all MIP global inversions agree that 2016 was the period that Australia recorded the largest carbon uptake, relative to the 2015-2018 mean. During the year 2016, five global inversions (PCTM, CMS-Flux, CT, OU and UT) agree with our findings and suggest November as the month of peak uptake.

In terms of carbon release, we observe that almost all OCO-2 MIP inversions agree that the largest outgassing occurred in November 2015. Our inversion places the maximum outgassing during the MIP study period in October 2017.

In summary, In summary, analysis of the inter-annual (peak-to-peak) variability shows that PCTM (3.05 PgC yr$^{-1}$) and CSU (2.46 PgC yr$^{-1}$) produce the largest amplitude of variability compared to our prior (0.70 PgC yr$^{-1}$) and posterior anomalies (1.89 PgC yr$^{-1}$). We also note that UO and AMES exhibit the lowest carbon amplitude of the variability, which values are

1.06 and 1.86 respectively. The disagreement between the global inversion and our study might also be driven by transport differences (Basu et al., 2018; Schuh et al., 2019).



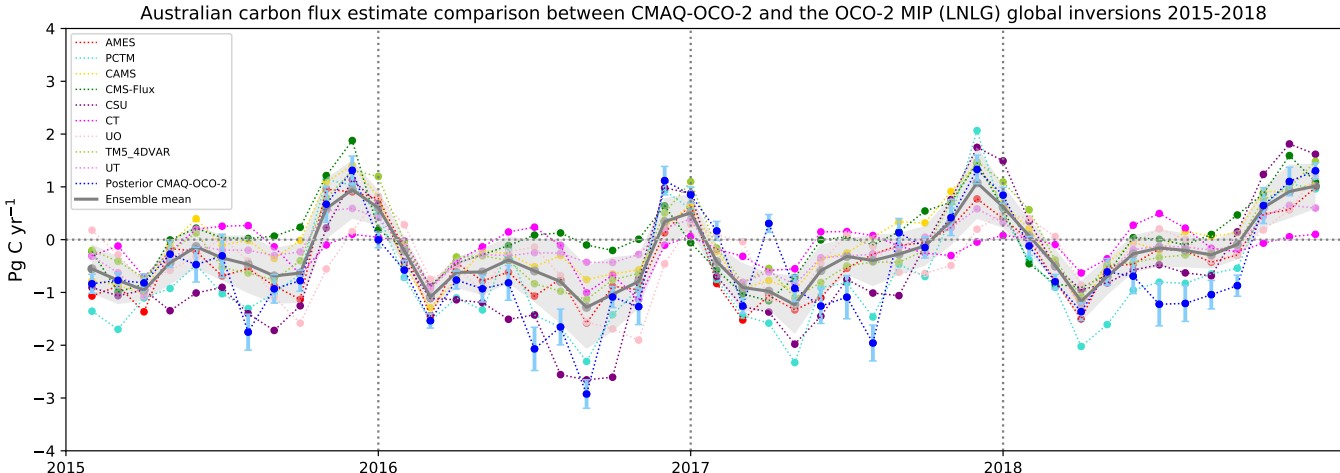

**Figure 17.** Comparison between 3-month running mean posterior (black line) and prior (grey line) terrestrial $CO_2$ flux anomalies (PgC $yr^{-1}$) and 3-month running mean anomalies of six different global transport models: AMES, PCTM, CAMS, CMS-Flux, CSU, CT, OU, TM5−4DVAR, UT

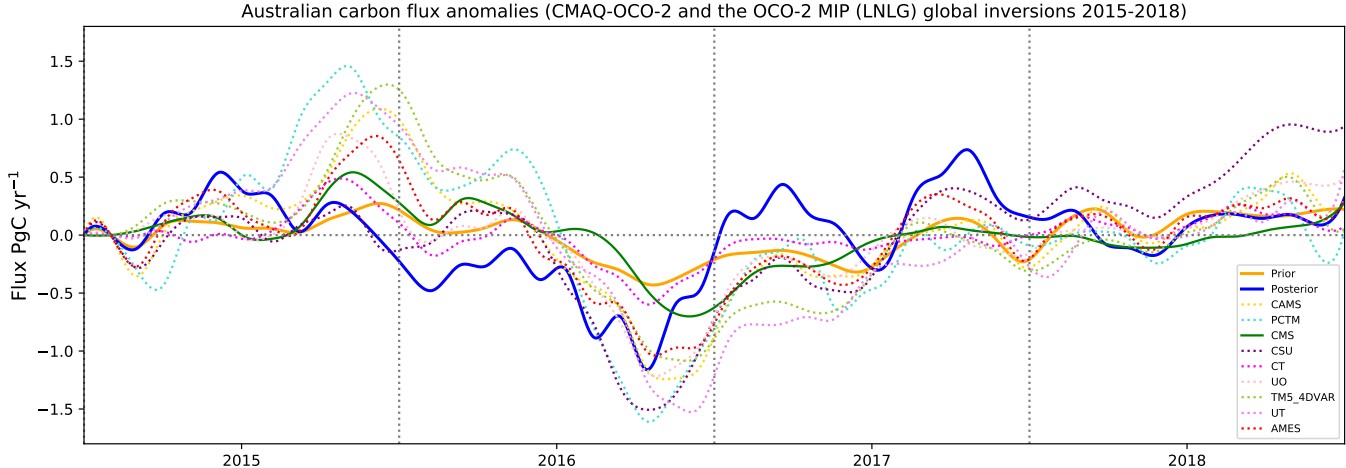

**Figure 18.** Comparison between 3-month running mean posterior (black line) and prior (grey line) terrestrial $CO_2$ flux anomalies (PgC $yr^{-1}$) and 3-month running mean anomalies of nine different global transport models: AMES, PCTM, CAMS, CMS-Flux, CSU, CT, OU, TM5−4DVAR, UT.

As a final discussion, we could say that previous inversion studies over Australia have been limited by the lack of in situ data. OCO-2 certainly allows a quantum leap in resolution but this is still reasonably coarse, especially when one remembers that the prior covariance structures we use impose smooth variations up to the correlation length of 500 km. Instruments with scanning

geometries which allow higher resolution observations such as OCO-3 (Eldering et al., 2019) may improve significantly the available resolution of fluxes. This is particularly important when assessing the roles of drivers such as rainfall which may vary





**Table 3.** Summary of the peak-to-peak amplitude of 3-month running mean posterior (black line) and prior (grey line) terrestrial carbon flux anomalies and 3-month running mean anomalies of nine different global transport models (Units PgC yr$^{-1}$).

| Carbon flux estimates | Models | Maximun | Date | Minimun | Date | Amplitud |
|---|---|---|---|---|---|---|
| CMAQ-OCO-2 inversion | Posterior | 0.78 | 2017-10-31 | -1.12 | 2016-10-31 | 1.89 |
| BIOS-CABLE3 | Prior | 0.20 | 2018-07-31 | -0.50 | 2016-11-30 | 0.70 |
| | AMES | 0.85 | 2015-12-31 | -1.01 | 2016-10-31 | 1.86 |
| | PCTM | 1.44 | 2015-11-30 | -1.61 | 2016-10-31 | 3.05 |
| | CAMS | 1.09 | 2015-12-31 | -1.23 | 2016-11-30 | 2.33 |
| | CMS-Flux | 0.54 | 2015-11-30 | -0.68 | 2016-12-31 | 1.23 |
| Global inversions | CSU | 0.95 | 2018-10-31 | -1.51 | 2016-10-31 | 2.46 |
| | CT | 0.46 | 2015-11-30 | -0.60 | 2016-10-31 | 1.06 |
| | OU | 0.83 | 2015-11-30 | -0.60 | 2016-10-31 | 1.43 |
| | TM5-4DVAR | 1.28 | 2015-12-31 | -1.19 | 2016-11-30 | 2.47 |
| | UT | 1.23 | 2015-11-30 | -1.07 | 2016-12-31 | 2.30 |

on smaller scales. We also note continuing improvement in the OCO retrievals themselves which should allow joint assimilation of land and ocean measurements, hopefully improving the visibility of coastal fluxes and improving comparison with coastal in situ measurements such as Cape grim and Gunn Point as shown by Villalobos et al. (2021).

## 5 Conclusions

We estimated monthly carbon fluxes over Australia for 2015 to 2019, based on the assimilation of the Orbiting Carbon Observatory-2 (OCO-2) satellite data (land nadir and glint data, version 9). We investigated the effect of vegetation productivity (EVI anomalies as proxy) and climate driver variations such as rainfall and air temperature on the Australian terrestrial carbon flux variability. The mean of our five-year inversion suggests that Australia was a carbon sink of -0.46 ± 0.08 PgC yr$^{-1}$ driven partly by large carbon uptake (-1.04 PgC yr$^{-1}$) recorded in 2016 over the savanna and sparsely vegetated ecosystems. We found that negative carbon flux anomalies recorded in this period over these ecosystems coincide with an increase in the vegetation greenness (positive EVI anomalies) driven by higher than average rainfall anomalies and lower than average air temperature anomalies. The 2017 sink over Australia also contributed to the 2015–2019 long term mean, but its contribution was not as significant as 2015 and 2016. Negative carbon flux anomalies recorded in 2017 also coincided with positive rainfall anomalies and temperatures below average in that period over areas with sparse vegetation. In 2018 we did not find significant terrestrial flux anomalies across Australia, and 2019 mainly was affected by positive carbon flux anomalies, which also were in line with a deficit of rainfall and positive temperature anomalies.

Regarding validation of our inversion with independent data, we found it challenging to validate our posterior column-averaged concentration with the current Australian monitoring sites. Despite the fact that for several periods between 2015-

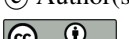



2019, the posterior concentration biases at the TCCON monitoring site were less than 1.0 ppm, OCO-2 data was not able to reduce prior biases significantly. We associate this slight or no improvement with the fact that these monitoring stations are strongly affected by ocean fluxes, where no OCO-2 data was considered. Similar findings were found for in-situ measurements at coastal sites such as Cape Grim and Gunn Point. Despite the weak comparison with independent monitoring data, the comparison to OCO-2 MIP global inversion for 2015-2018 present similar results to our regional inversion, suggesting that the

year 2016 was a period in which Australia acted as a strong carbon sink of $CO_2$.





## Appendix A:  Spatial pattern of OCO-2 soundings across Australia

**Figure A1.** Spatial distribution of OCO-2 soundings (Land nadir and glint data) over the CMAQ domain for 2015.

**Figure A2.** Spatial distribution of OCO-2 soundings (Land nadir and glint data) over the CMAQ domain for 2016.





**Figure A3.** Spatial distribution of OCO-2 soundings (Land nadir and glint data) over the CMAQ domain for 2017.





**Figure A4.** Spatial distribution of OCO-2 soundings (Land nadir and glint data) over the CMAQ domain for 2018.



**Figure A5.** Spatial distribution of OCO-2 soundings (Land nadir and glint data) over the CMAQ domain for 2019.





## Appendix B: Analysis of the residual TCCON

**Table B1.** Analysis of the residual between CMAQ prior and posterior simulation and TCCON Darwin site for 2015–2019. Averaged bias (Bias), Root-mean-square error (RMSE) and Pearson's coefficient (R).

| | | Darwin | | | | | | | | | | | | | | | | | | | | | |
|---|---|---|---|---|---|---|---|---|---|---|---|---|---|---|---|---|---|---|---|---|---|---|---|
| months | Prior | | | Posterior | | | months | Prior | | | Posterior | | | months | Prior | | | Posterior | | | | | |
| yyyy-mm | Bias | RMSE | R | Bias | RMSE | R | yyyy-mm | Bias | RMSE | R | Bias | RMSE | R | yyyy-mm | Bias | RMSE | R | Bias | RMSE | R | | | |
| 2015-01 | 0.12 | 0.51 | 0.81 | -0.04 | 0.82 | 0.75 | 2017-01 | - | - | - | - | - | - | 2019-01 | - | - | - | - | - | - | | | |
| 2015-02 | 0.69 | 0.85 | 0.78 | 0.38 | 0.63 | 0.78 | 2017-02 | - | - | - | - | - | - | 2019-02 | -1.31 | 1.34 | -0.49 | -1.40 | 1.43 | -0.61 | | | |
| 2015-03 | 0.93 | 1.10 | 0.14 | 0.18 | 0.59 | 0.29 | 2017-03 | 0.32 | 0.58 | 0.60 | -1.34 | 1.45 | 0.59 | 2019-03 | -0.78 | 0.87 | 0.60 | -1.42 | 1.55 | 0.39 | | | |
| 2015-04 | 0.85 | 0.94 | 0.38 | 0.60 | 0.74 | 0.42 | 2017-04 | - | - | - | - | - | - | 2019-04 | - | - | - | - | - | - | | | |
| 2015-05 | 0.97 | 1.05 | 0.37 | 0.90 | 0.99 | 0.52 | 2017-05 | - | - | - | - | - | - | 2019-05 | - | - | - | - | - | - | | | |
| 2015-06 | 0.90 | 0.97 | 0.21 | 1.24 | 1.27 | 0.23 | 2017-06 | -0.12 | 0.44 | -0.02 | 0.24 | 0.46 | 0.25 | 2019-06 | - | - | - | - | - | - | | | |
| 2015-07 | 1.51 | 1.55 | -0.18 | 1.07 | 1.10 | 0.22 | 2017-07 | -0.13 | 0.74 | 0.07 | 0.04 | 0.69 | 0.16 | 2019-07 | - | - | - | - | - | - | | | |
| 2015-08 | 1.44 | 1.46 | 0.34 | 1.06 | 1.10 | 0.35 | 2017-08 | 0.19 | 0.39 | 0.09 | 0.28 | 0.49 | 0.15 | 2019-08 | - | - | - | - | - | - | | | |
| 2015-09 | 1.12 | 1.16 | 0.02 | 0.81 | 0.86 | 0.10 | 2017-09 | -0.15 | 0.32 | 0.14 | -0.16 | 0.31 | 0.13 | 2019-09 | -0.30 | 0.66 | 0.25 | -0.40 | 0.73 | 0.18 | | | |
| 2015-10 | 0.55 | 0.63 | 0.53 | 0.63 | 0.69 | 0.62 | 2017-10 | -0.92 | 1.05 | 0.42 | -0.51 | 0.72 | 0.46 | 2019-10 | -0.91 | 1.01 | 0.72 | -0.87 | 0.95 | 0.78 | | | |
| 2015-11 | -0.25 | 0.51 | 0.66 | 0.11 | 0.42 | 0.75 | 2017-11 | -1.12 | 1.21 | 0.06 | -0.76 | 0.99 | -0.20 | 2019-11 | -1.31 | 1.37 | 0.70 | -0.96 | 1.06 | 0.61 | | | |
| 2015-12 | -0.34 | 0.48 | 0.18 | -0.02 | 0.31 | 0.26 | 2017-12 | -0.67 | 0.79 | 0.37 | -0.74 | 0.97 | 0.26 | 2019-12 | -1.37 | 1.43 | 0.35 | -1.22 | 1.32 | 0.26 | | | |
| 2016-01 | 0.27 | 0.55 | 0.39 | 0.09 | 0.60 | 0.28 | 2018-01 | -0.57 | 0.85 | 0.08 | -1.15 | 1.30 | -0.08 | | | | | | | | | | |
| 2016-02 | 0.34 | 0.63 | 0.34 | -0.25 | 0.66 | 0.29 | 2018-02 | -0.90 | 1.04 | 0.08 | -1.98 | 2.11 | -0.13 | | | | | | | | | | |
| 2016-03 | 0.44 | 0.63 | 0.36 | -0.09 | 0.58 | 0.30 | 2018-03 | -0.62 | 1.06 | -0.65 | -1.46 | 1.67 | -0.59 | | | | | | | | | | |
| 2016-04 | 0.80 | 0.93 | 0.34 | 0.86 | 0.95 | 0.48 | 2018-04 | - | - | - | - | - | - | | | | | | | | | | |
| 2016-05 | 0.26 | 0.41 | 0.58 | 0.41 | 0.53 | 0.54 | 2018-05 | - | - | - | - | - | - | | | | | | | | | | |
| 2016-06 | 0.37 | 0.45 | 0.16 | 0.21 | 0.38 | 0.18 | 2018-06 | -0.27 | 0.51 | -0.08 | -0.09 | 0.40 | 0.22 | | | | | | | | | | |
| 2016-07 | 0.18 | 0.42 | 0.17 | 0.07 | 0.38 | 0.34 | 2018-07 | -0.34 | 0.52 | -0.16 | -0.34 | 0.48 | 0.04 | | | | | | | | | | |
| 2016-08 | 0.09 | 0.33 | 0.04 | -0.07 | 0.33 | 0.27 | 2018-08 | -0.34 | 0.49 | 0.05 | -0.94 | 0.99 | 0.07 | | | | | | | | | | |
| 2016-09 | -0.17 | 0.40 | 0.25 | -0.20 | 0.40 | 0.26 | 2018-09 | -0.57 | 0.69 | 0.09 | -0.90 | 0.99 | 0.09 | | | | | | | | | | |
| 2016-10 | -0.31 | 0.42 | 0.01 | -0.30 | 0.47 | 0.27 | 2018-10 | - | - | - | - | - | - | | | | | | | | | | |
| 2016-11 | -0.40 | 0.42 | 0.75 | -0.43 | 0.47 | 0.67 | 2018-11 | - | - | - | - | - | - | | | | | | | | | | |
| 2016-12 | - | - | - | - | - | - | 2018-12 | - | - | - | - | - | - | | | | | | | | | | |





**Table B2.** Analysis of the residual between CMAQ prior and posterior simulation and TCCON Lauder site for 2015–2019. Averaged bias (Bias), Root-mean-square error (RMSE) and Pearson's coefficient (R).

| | | | | | | | | | | | | | Lauder | | | | | | | | |
|---|---|---|---|---|---|---|---|---|---|---|---|---|---|---|---|---|---|---|---|---|---|
| Months | Prior | | | Posterior | | | Months | Prior | | | Posterior | | | Months | Prior | | | Posterior | | | |
| yyyy-mm | Bias | RMSE | R | Bias | RMSE | R | yyyy-mm | Bias | RMSE | R | Bias | RMSE | R | yyyy-mm | Bias | RMSE | R | Bias | RMSE | R | |
| 2015-01 | 0.48 | 0.58 | 0.31 | 0.71 | 0.85 | 0.06 | 2017-01 | 0.571 | 0.71 | 0.39 | 0.60 | 0.79 | 0.39 | 2019-01 | 0.89 | 1.01 | 0.53 | 1.10 | 1.38 | 0.55 | |
| 2015-02 | 0.61 | 0.74 | 0.22 | 1.03 | 1.17 | 0.34 | 2017-02 | 0.187 | 0.41 | 0.24 | 0.65 | 1.03 | 0.07 | 2019-02 | 0.94 | 1.02 | 0.21 | 1.10 | 1.40 | 0.12 | |
| 2015-03 | 0.54 | 0.62 | 0.51 | 0.73 | 0.84 | 0.53 | 2017-03 | 0.16 | 0.41 | 0.35 | 0.27 | 0.48 | 0.35 | 2019-03 | 0.64 | 0.75 | 0.53 | 0.95 | 1.25 | 0.69 | |
| 2015-04 | 0.50 | 0.59 | 0.77 | 0.51 | 0.60 | 0.79 | 2017-04 | 0.35 | 0.52 | 0.22 | 0.20 | 0.32 | 0.21 | 2019-04 | 0.83 | 0.88 | 0.58 | 0.80 | 0.94 | 0.54 | |
| 2015-05 | 0.82 | 0.89 | 0.30 | 0.83 | 0.90 | 0.23 | 2017-05 | 0.191 | 0.40 | 0.54 | 0.16 | 0.24 | 0.59 | 2019-05 | 0.73 | 0.81 | 0.65 | 0.62 | 0.75 | 0.67 | |
| 2015-06 | 0.65 | 0.86 | 0.60 | 0.61 | 0.82 | 0.56 | 2017-06 | 0.44 | 0.58 | 0.77 | 0.32 | 0.46 | 0.76 | 2019-06 | 0.63 | 0.76 | 0.85 | 0.46 | 0.68 | 0.82 | |
| 2015-07 | 0.69 | 0.82 | 0.79 | 0.64 | 0.79 | 0.76 | 2017-07 | 0.44 | 0.56 | 0.30 | 0.26 | 0.39 | 0.38 | 2019-07 | 0.81 | 0.88 | 0.73 | 0.65 | 0.80 | 0.69 | |
| 2015-08 | 0.57 | 0.64 | 0.64 | 0.57 | 0.64 | 0.66 | 2017-08 | 0.71 | 0.81 | 0.64 | 0.63 | 0.87 | 0.63 | 2019-08 | 1.14 | 1.22 | 0.75 | 1.15 | 1.42 | 0.74 | |
| 2015-09 | 0.71 | 0.73 | 0.83 | 0.63 | 0.67 | 0.83 | 2017-09 | 1.30 | 1.35 | 0.73 | 1.58 | 1.76 | 0.71 | 2019-09 | 0.96 | 1.07 | 0.60 | 0.97 | 1.24 | 0.63 | |
| 2015-10 | 0.75 | 0.82 | 0.65 | 0.74 | 0.82 | 0.59 | 2017-10 | 1.01 | 1.08 | 0.72 | 1.28 | 1.51 | 0.76 | 2019-10 | 1.09 | 1.13 | 0.72 | 1.23 | 1.42 | 0.62 | |
| 2015-11 | 0.52 | 0.72 | 0.36 | 0.43 | 0.65 | 0.37 | 2017-11 | 0.74 | 0.90 | 0.50 | 0.79 | 0.97 | 0.50 | 2019-11 | 0.99 | 1.06 | 0.79 | 1.09 | 1.33 | 0.78 | |
| 2015-12 | 0.71 | 0.76 | 0.79 | 0.77 | 0.81 | 0.81 | 2017-12 | 0.83 | 0.91 | 0.63 | 1.37 | 1.65 | 0.67 | 2019-12 | 1.03 | 1.10 | 0.70 | 0.82 | 1.11 | 0.52 | |
| 2016-01 | 0.43 | 0.51 | 0.81 | 0.40 | 0.51 | 0.78 | 2018-01 | 0.64 | 0.73 | 0.33 | 1.04 | 1.43 | 0.16 | | | | | | | | |
| 2016-02 | 0.23 | 0.40 | 0.54 | 0.18 | 0.33 | 0.49 | 2018-02 | 0.52 | 0.61 | 0.62 | 0.46 | 0.62 | 0.63 | | | | | | | | |
| 2016-03 | 0.24 | 0.45 | 0.57 | 0.23 | 0.32 | 0.53 | 2018-03 | 0.59 | 0.74 | 0.35 | 0.52 | 0.83 | 0.33 | | | | | | | | |
| 2016-04 | 0.17 | 0.45 | 0.72 | 0.19 | 0.25 | 0.71 | 2018-04 | 0.63 | 0.74 | 0.44 | 0.53 | 0.71 | 0.42 | | | | | | | | |
| 2016-05 | 0.39 | 0.54 | 0.61 | 0.29 | 0.46 | 0.55 | 2018-05 | 0.91 | 1.04 | 0.40 | 0.99 | 1.46 | 0.39 | | | | | | | | |
| 2016-06 | 0.21 | 0.50 | 0.44 | 0.22 | 0.38 | 0.48 | 2018-06 | 0.70 | 0.89 | -0.42 | 0.68 | 0.98 | -0.44 | | | | | | | | |
| 2016-07 | 0.59 | 0.78 | 0.74 | 0.58 | 0.82 | 0.74 | 2018-07 | 0.99 | 1.03 | 0.69 | 0.88 | 1.13 | 0.43 | | | | | | | | |
| 2016-08 | 0.32 | 0.55 | 0.55 | 0.30 | 0.41 | 0.55 | 2018-08 | 0.75 | 0.79 | 0.77 | 0.60 | 0.72 | 0.78 | | | | | | | | |
| 2016-09 | 0.31 | 0.62 | 0.11 | 0.34 | 0.62 | 0.16 | 2018-09 | 0.96 | 1.03 | 0.41 | 0.94 | 1.16 | 0.45 | | | | | | | | |
| 2016-10 | 0.30 | 0.55 | 0.10 | 0.58 | 0.75 | 0.37 | 2018-10 | 1.011 | 1.11 | 0.48 | 1.34 | 1.58 | 0.55 | | | | | | | | |
| 2016-11 | 0.93 | 0.97 | 0.68 | 1.30 | 1.47 | 0.66 | 2018-11 | 1.25 | 1.30 | 0.63 | 2.10 | 2.31 | 0.65 | | | | | | | | |
| 2016-12 | 0.576 | 0.7823 | 0.22 | 1.16 | 1.78 | 0.16 | 2018-12 | 1.35 | 1.40 | 0.33 | 2.897 | 3.15 | 0.33 | | | | | | | | |





**Table B3.** Analysis of the residual between CMAQ prior and posterior simulation and TCCON Wollongong site for 2015–2019. Averaged bias (Bias), Root-mean-square error (RMSE) and Pearson's coefficient (R).

| | Wollongong | | | | | | | | | | | | | | | | | | | | |
|---|---|---|---|---|---|---|---|---|---|---|---|---|---|---|---|---|---|---|---|---|
| Months | Prior | | | Posterior | | | Months | Prior | | | Posterior | | | Months | Prior | | | Posterior | | |
| yyyy-mm | Bias | RMSE | R | Bias | RMSE | R | yyyy-mm | Bias | RMSE | R | Bias | RMSE | R | yyyy-mm | Bias | RMSE | R | Bias | RMSE | R |
| 2015-01 | -0.04 | 0.72 | 0.21 | 0.07 | 0.75 | 0.23 | 2017-01 | -0.76 | 0.99 | 0.25 | -0.35 | 0.84 | 0.16 | 2019-01 | -2.20 | 2.55 | 0.07 | -2.23 | 2.51 | 0.20 |
| 2015-02 | -0.21 | 0.56 | 0.48 | 0.16 | 0.63 | 0.51 | 2017-02 | -0.79 | 1.08 | 0.37 | -0.82 | 1.13 | 0.14 | 2019-02 | -0.47 | 0.84 | 0.28 | -0.65 | 0.93 | 0.32 |
| 2015-03 | 0.66 | 0.94 | 0.19 | 0.51 | 0.88 | 0.16 | 2017-03 | -0.83 | 1.26 | -0.07 | -0.82 | 1.32 | 0.01 | 2019-03 | -0.33 | 0.65 | 0.48 | -0.51 | 0.78 | 0.45 |
| 2015-04 | 0.72 | 0.96 | 0.07 | 0.82 | 1.06 | 0.15 | 2017-04 | -0.52 | 1.11 | 0.03 | -0.55 | 1.13 | -0.09 | 2019-04 | -0.50 | 0.83 | 0.16 | -0.65 | 0.95 | 0.16 |
| 2015-05 | 1.26 | 1.40 | 0.12 | 1.54 | 1.72 | 0.02 | 2017-05 | -0.42 | 0.66 | 0.31 | -0.50 | 0.74 | 0.26 | 2019-05 | 0.03 | 0.67 | 0.26 | -0.06 | 0.71 | 0.21 |
| 2015-06 | 1.41 | 1.53 | 0.68 | 1.61 | 1.72 | 0.68 | 2017-06 | -0.24 | 0.53 | 0.67 | -0.24 | 0.53 | 0.68 | 2019-06 | 0.20 | 0.75 | 0.17 | -0.14 | 0.60 | 0.40 |
| 2015-07 | 1.37 | 1.56 | 0.32 | 1.14 | 1.38 | 0.28 | 2017-07 | -0.01 | 0.55 | 0.42 | -0.11 | 0.59 | 0.37 | 2019-07 | 0.17 | 0.70 | 0.17 | -0.04 | 0.62 | 0.17 |
| 2015-08 | 1.42 | 1.57 | 0.25 | 1.61 | 1.76 | 0.28 | 2017-08 | 0.17 | 0.89 | -0.19 | 0.16 | 0.91 | -0.19 | 2019-08 | -0.29 | 0.52 | 0.82 | -0.77 | 0.87 | 0.79 |
| 2015-09 | 1.19 | 1.44 | 0.16 | 1.11 | 1.42 | 0.19 | 2017-09 | 0.31 | 0.86 | -0.06 | 0.17 | 0.68 | 0.12 | 2019-09 | -0.05 | 0.67 | -0.10 | -0.67 | 0.98 | -0.18 |
| 2015-10 | 0.07 | 0.72 | 0.03 | 0.29 | 0.83 | 0.00 | 2017-10 | - | - | - | - | - | - | 2019-10 | -0.36 | 0.64 | 0.38 | -0.74 | 0.97 | 0.22 |
| 2015-11 | -0.74 | 1.22 | -0.08 | -0.40 | 1.13 | -0.05 | 2017-11 | - | - | - | - | - | - | 2019-11 | -0.14 | 0.80 | 0.62 | -0.63 | 0.95 | 0.69 |
| 2015-12 | -0.45 | 0.69 | 0.14 | -0.60 | 0.85 | -0.03 | 2017-12 | - | - | - | - | - | - | 2019-12 | 0.99 | 1.77 | 0.67 | -1.31 | 1.44 | 0.69 |
| 2016-01 | -0.29 | 0.51 | 0.58 | -0.19 | 0.42 | 0.50 | 2018-01 | - | - | - | - | - | - | | | | | | | |
| 2016-02 | -0.93 | 1.09 | 0.24 | -1.00 | 1.18 | -0.13 | 2018-02 | - | - | - | - | - | - | | | | | | | |
| 2016-03 | -0.18 | 0.56 | 0.59 | -0.36 | 0.69 | 0.54 | 2018-03 | -0.72 | 0.94 | 0.47 | -0.98 | 1.15 | 0.30 | | | | | | | |
| 2016-04 | 0.10 | 0.69 | -0.20 | 0.07 | 0.57 | -0.19 | 2018-04 | -0.221 | 0.66 | 0.52 | -0.59 | 0.78 | 0.55 | | | | | | | |
| 2016-05 | -0.07 | 0.65 | 0.31 | -0.12 | 0.71 | 0.17 | 2018-05 | -0.011 | 0.59 | 0.20 | -0.17 | 0.58 | 0.11 | | | | | | | |
| 2016-06 | -0.21 | 0.71 | -0.13 | -0.52 | 0.87 | -0.16 | 2018-06 | -0.03 | 0.39 | 0.72 | -0.11 | 0.45 | 0.66 | | | | | | | |
| 2016-07 | 0.04 | 0.66 | 0.52 | -0.05 | 0.71 | 0.49 | 2018-07 | -0.15 | 0.65 | 0.21 | -0.33 | 0.68 | 0.19 | | | | | | | |
| 2016-08 | 0.48 | 0.80 | 0.17 | 0.07 | 0.70 | 0.22 | 2018-08 | 0.08 | 0.51 | 0.42 | 0.23 | 0.57 | 0.37 | | | | | | | |
| 2016-09 | -0.05 | 0.89 | 0.40 | -0.37 | 1.05 | 0.36 | 2018-09 | -0.25 | 0.78 | 0.13 | -0.49 | 0.91 | 0.03 | | | | | | | |
| 2016-10 | -0.14 | 0.78 | 0.17 | -0.16 | 0.86 | 0.16 | 2018-10 | -0.352 | 0.78 | -0.08 | -0.64 | 1.00 | -0.22 | | | | | | | |
| 2016-11 | -0.53 | 0.93 | 0.30 | 0.41 | 0.87 | 0.32 | 2018-11 | -0.808 | 1.05 | 0.64 | -0.29 | 0.93 | 0.52 | | | | | | | |
| 2016-12 | -0.77 | 1.1305 | 0.06 | -0.13 | 0.87 | 0.07 | 2018-12 | -1.10 | 1.31 | 0.66 | -0.81 | 1.10 | 0.75 | | | | | | | |





# Appendix C: Analysis of the residual in-situ data

**Table C1.** Analysis of the residual between CMAQ prior and posterior simulation and Cape Grim site for 2015–2019. Averaged bias (Bias), Root-mean-square error (RMSE) and Pearson's coefficient (R).

| | | | | | | | Cape Grim | | | | | | | | | | | | | | |
|---|---|---|---|---|---|---|---|---|---|---|---|---|---|---|---|---|---|---|---|---|
| months | Prior | | | Posterior | | | months | Prior | | | Posterior | | | months | Prior | | | Posterior | | |
| yyyy-mm | Bias | RMSE | R | Bias | RMSE | R | yyyy-mm | Bias | RMSE | R | Bias | RMSE | R | yyyy-mm | Bias | RMSE | R | Bias | RMSE | R |
| 2015-01 | -2.31 | 3.38 | 0.28 | -2.33 | 3.22 | 0.28 | 2017-01 | -2.27 | 3.15 | 0.01 | -2.04 | 2.48 | 0.59 | 2019-01 | -1.81 | 3.15 | -0.11 | -2.99 | 3.65 | 0.07 |
| 2015-02 | -2.65 | 3.68 | 0.57 | -2.59 | 3.91 | 0.53 | 2017-02 | -2.08 | 2.93 | -0.11 | -1.99 | 2.53 | 0.40 | 2019-02 | -1.13 | 3.06 | -0.14 | -2.22 | 2.98 | 0.31 |
| 2015-03 | -1.25 | 2.02 | 0.53 | -2.70 | 3.07 | 0.29 | 2017-03 | -1.70 | 4.22 | 0.17 | -2.60 | 4.57 | 0.24 | 2019-03 | -1.67 | 3.33 | 0.37 | -2.51 | 3.22 | 0.60 |
| 2015-04 | -2.33 | 3.22 | 0.41 | -2.16 | 3.54 | 0.19 | 2017-04 | -2.23 | 3.21 | 0.58 | -1.95 | 3.25 | 0.45 | 2019-04 | -1.60 | 3.46 | -0.24 | -2.79 | 3.41 | 0.49 |
| 2015-05 | -1.85 | 3.27 | 0.36 | -0.60 | 2.82 | 0.46 | 2017-05 | -2.22 | 4.21 | 0.06 | -2.71 | 3.95 | 0.30 | 2019-05 | -2.46 | 3.64 | 0.22 | -2.60 | 3.46 | 0.46 |
| 2015-06 | -1.77 | 2.79 | 0.14 | -0.88 | 2.28 | 0.20 | 2017-06 | -1.77 | 2.81 | -0.01 | -2.17 | 2.88 | 0.21 | 2019-06 | -2.31 | 3.81 | -0.07 | -6.70 | 8.35 | -0.59 |
| 2015-07 | -0.96 | 2.05 | 0.10 | -2.18 | 3.20 | -0.03 | 2017-07 | -1.64 | 2.71 | -0.06 | -3.01 | 3.65 | 0.30 | 2019-07 | -1.50 | 2.37 | -0.15 | -2.80 | 3.18 | 0.30 |
| 2015-08 | -1.91 | 2.93 | -0.05 | -2.12 | 3.22 | 0.02 | 2017-08 | -1.66 | 2.37 | 0.09 | -1.82 | 2.46 | 0.01 | 2019-08 | -1.64 | 2.45 | -0.17 | -4.35 | 4.98 | 0.21 |
| 2015-09 | -2.29 | 3.63 | -0.02 | -4.18 | 4.94 | 0.16 | 2017-09 | -1.08 | 2.02 | -0.02 | -1.55 | 2.17 | 0.10 | 2019-09 | -2.23 | 2.95 | 0.19 | -4.35 | 4.97 | -0.31 |
| 2015-10 | -2.34 | 3.44 | 0.08 | -2.52 | 3.75 | 0.00 | 2017-10 | -2.03 | 2.93 | 0.03 | -3.26 | 4.07 | -0.14 | 2019-10 | -2.58 | 4.07 | 0.13 | -4.71 | 6.14 | -0.18 |
| 2015-11 | -2.35 | 3.32 | 0.34 | -3.07 | 4.28 | -0.06 | 2017-11 | -1.60 | 3.09 | 0.09 | 0.76 | 3.38 | 0.52 | 2019-11 | -1.81 | 2.94 | 0.52 | -2.14 | 3.61 | -0.05 |
| 2015-12 | -1.86 | 2.51 | 0.58 | -2.34 | 2.89 | 0.49 | 2017-12 | -1.88 | 2.51 | 0.07 | -1.77 | 2.44 | 0.38 | 2019-12 | -1.31 | 3.86 | -0.07 | -5.14 | 5.76 | 0.05 |
| 2016-01 | -1.53 | 2.44 | 0.60 | -2.40 | 3.01 | 0.55 | 2018-01 | -2.23 | 3.96 | 0.05 | -2.88 | 3.66 | 0.46 | | | | | | | |
| 2016-02 | -2.28 | 3.49 | 0.35 | -3.65 | 4.78 | 0.06 | 2018-02 | -1.72 | 3.92 | -0.08 | -3.52 | 4.20 | 0.48 | | | | | | | |
| 2016-03 | -1.64 | 2.79 | 0.43 | -3.54 | 4.25 | 0.19 | 2018-03 | -1.80 | 2.88 | 0.05 | -2.95 | 3.38 | 0.16 | | | | | | | |
| 2016-04 | -2.15 | 3.22 | 0.57 | -3.26 | 4.20 | 0.40 | 2018-04 | -1.85 | 3.71 | 0.05 | -3.48 | 4.36 | 0.20 | | | | | | | |
| 2016-05 | -1.62 | 2.51 | 0.68 | -1.56 | 2.60 | 0.61 | 2018-05 | -1.55 | 2.74 | 0.22 | -3.14 | 3.91 | 0.01 | | | | | | | |
| 2016-06 | -2.26 | 3.23 | 0.28 | -2.96 | 3.80 | 0.28 | 2018-06 | -1.89 | 3.18 | 0.07 | -2.03 | 3.04 | 0.27 | | | | | | | |
| 2016-07 | -1.22 | 2.07 | 0.52 | -1.64 | 2.32 | 0.55 | 2018-07 | -1.38 | 2.28 | -0.06 | -2.59 | 3.19 | 0.01 | | | | | | | |
| 2016-08 | -1.42 | 2.74 | 0.31 | -3.24 | 3.90 | 0.60 | 2018-08 | -1.35 | 2.11 | 0.05 | -1.37 | 2.23 | 0.29 | | | | | | | |
| 2016-09 | -1.43 | 2.61 | 0.24 | -1.60 | 2.77 | 0.22 | 2018-09 | -1.69 | 2.79 | -0.16 | -2.44 | 3.08 | 0.15 | | | | | | | |
| 2016-10 | -2.36 | 2.82 | 0.48 | -2.22 | 2.79 | 0.48 | 2018-10 | -1.61 | 2.70 | 0.13 | -2.94 | 3.46 | 0.31 | | | | | | | |
| 2016-11 | -2.79 | 3.47 | 0.03 | -1.89 | 2.50 | 0.56 | 2018-11 | -1.97 | 3.35 | 0.15 | -0.85 | 3.17 | 0.30 | | | | | | | |
| 2016-12 | -2.22 | 3.26 | 0.30 | -1.15 | 2.96 | 0.38 | 2018-12 | -2.05 | 3.53 | -0.20 | -1.17 | 2.78 | 0.36 | | | | | | | |





**Table C2.** Analysis of the residual between CMAQ prior and posterior simulation and Gunn Point site for 2015–2019. Averaged bias (Bias), Root-mean-square error (RMSE) and Pearson's coefficient (R).

| | | | | | | | Gunn Point | | | | | | | | | | | | | | |
|---|---|---|---|---|---|---|---|---|---|---|---|---|---|---|---|---|---|---|---|---|
| months | Prior | | | Posterior | | | months | Prior | | | Posterior | | | months | Prior | | | Posterior | | |
| yyyy-mm | Bias | RMSE | R | Bias | RMSE | R | yyyy-mm | Bias | RMSE | R | Bias | RMSE | R | yyyy-mm | Bias | RMSE | R | Bias | RMSE | R |
| 2015-01 | -1.16 | 4.83 | 0.37 | -2.11 | 4.87 | 0.26 | 2017-01 | - | - | - | - | - | - | 2019-01 | -3.19 | 5.56 | -0.20 | -7.19 | 9.22 | -0.07 |
| 2015-02 | -2.88 | 4.73 | 0.41 | -3.55 | 4.07 | 0.47 | 2017-02 | - | - | - | - | - | - | 2019-02 | -5.04 | 5.80 | 0.24 | -6.25 | 7.12 | 0.28 |
| 2015-03 | -1.93 | 4.21 | -0.06 | -3.36 | 4.84 | -0.06 | 2017-03 | - | - | - | - | - | - | 2019-03 | -2.99 | 4.37 | 0.37 | -6.36 | 7.13 | 0.30 |
| 2015-04 | -1.07 | 2.92 | 0.33 | -2.74 | 3.44 | 0.28 | 2017-04 | - | - | - | - | - | - | 2019-04 | -0.27 | 4.03 | 0.27 | 2.12 | 5.48 | 0.29 |
| 2015-05 | -1.76 | 2.78 | 0.35 | -3.65 | 3.38 | 0.53 | 2017-05 | - | - | - | - | - | - | 2019-05 | -1.40 | 3.42 | 0.45 | 0.11 | 4.24 | 0.60 |
| 2015-06 | -0.96 | 1.68 | 0.29 | 1.90 | 4.07 | 0.31 | 2017-06 | - | - | - | - | - | - | 2019-06 | -0.23 | 1.75 | 0.45 | -4.04 | 3.70 | 0.57 |
| 2015-07 | -2.53 | 16.46 | 0.00 | -7.71 | 17.67 | 0.06 | 2017-07 | - | - | - | - | - | - | 2019-07 | 1.69 | 2.65 | 0.01 | -2.12 | 3.78 | 0.20 |
| 2015-08 | 1.70 | 2.43 | 0.41 | -2.88 | 4.21 | 0.25 | 2017-08 | - | - | - | - | - | - | 2019-08 | 1.17 | 4.47 | -0.03 | -1.75 | 4.59 | -0.12 |
| 2015-09 | 1.81 | 2.13 | 0.28 | -0.32 | 1.54 | 0.04 | 2017-09 | - | - | - | - | - | - | 2019-09 | 2.69 | 3.02 | 0.23 | 1.08 | 1.42 | 0.47 |
| 2015-10 | 2.19 | 2.44 | 0.15 | 3.24 | 3.84 | -0.03 | 2017-10 | - | - | - | - | - | - | 2019-10 | 2.31 | 3.25 | -0.30 | 3.51 | 4.28 | -0.01 |
| 2015-11 | -0.52 | 2.30 | -0.67 | 0.66 | 2.63 | -0.63 | 2017-11 | - | - | - | - | - | - | 2019-11 | -0.63 | 1.99 | -0.04 | 2.21 | 2.52 | 0.18 |
| 2015-12 | -2.69 | 3.34 | 0.38 | -4.03 | 4.45 | 0.36 | 2017-12 | - | - | - | - | - | - | 2019-12 | -1.36 | 2.11 | 0.15 | -0.42 | 3.96 | 0.03 |
| 2016-01 | -3.70 | 4.66 | 0.16 | -3.74 | 3.95 | 0.16 | 2018-01 | - | - | - | - | - | - | | | | | | | |
| 2016-02 | -3.40 | 4.71 | 0.33 | -5.19 | 6.22 | 0.36 | 2018-02 | - | - | - | - | - | - | | | | | | | |
| 2016-03 | -2.74 | 4.03 | -0.02 | -3.30 | 4.30 | 0.02 | 2018-03 | - | - | - | - | - | - | | | | | | | |
| 2016-04 | 1.21 | 3.14 | -0.38 | 1.92 | 3.12 | -0.03 | 2018-04 | - | - | - | - | - | - | | | | | | | |
| 2016-05 | -1.69 | 4.84 | -0.27 | -0.50 | 4.45 | 0.02 | 2018-05 | - | - | - | - | - | - | | | | | | | |
| 2016-06 | -2.91 | 3.12 | 0.68 | -4.28 | 4.02 | 0.79 | 2018-06 | - | - | - | - | - | - | | | | | | | |
| 2016-07 | - | - | - | - | - | - | 2018-07 | - | - | - | - | - | - | | | | | | | |
| 2016-08 | - | - | - | - | - | - | 2018-08 | 0.51 | 1.61 | 0.47 | -6.16 | 5.70 | -0.29 | | | | | | | |
| 2016-09 | - | - | - | - | - | - | 2018-09 | 0.83 | 1.37 | 0.15 | -1.95 | 1.65 | 0.01 | | | | | | | |
| 2016-10 | - | - | - | - | - | - | 2018-10 | 0.09 | 0.76 | 0.39 | 0.89 | 0.92 | 0.77 | | | | | | | |
| 2016-11 | - | - | - | - | - | - | 2018-11 | -4.24 | 4.63 | -0.11 | -4.08 | 5.45 | 0.05 | | | | | | | |
| 2016-12 | - | - | - | - | - | - | 2018-12 | -3.63 | 4.60 | 0.02 | -4.70 | 5.88 | 0.31 | | | | | | | |





**Table C3.** Analysis of the residual between CMAQ prior and posterior simulation and Iron bark site for 2015–2019. Averaged bias (Bias), Root-mean-square error (RMSE) and Pearson's coefficient (R).

| | | | | | | | Iron Bark | | | | | | | | | | | | |
|---|---|---|---|---|---|---|---|---|---|---|---|---|---|---|---|---|---|---|---|
| months | Prior | | | Posterior | | | months | Prior | | | Posterior | | | months | Prior | | | Posterior | | |
| yyyy-mm | Bias | RMSE | R | Bias | RMSE | R | yyyy-mm | Bias | RMSE | R | Bias | RMSE | R | yyyy-mm | Bias | RMSE | R | Bias | RMSE | R |
| 2015-01 | -1.61 | 2.28 | 0.32 | 0.43 | 2.33 | 0.23 | 2017-01 | -1.02 | 1.09 | 0.77 | 0.82 | 0.90 | 0.01 | 2019-01 | - | - | - | - | - | - |
| 2015-02 | -1.07 | 1.30 | 0.44 | 0.71 | 1.14 | 0.06 | 2017-02 | - | - | - | - | - | - | 2019-02 | - | - | - | - | - | - |
| 2015-03 | -0.34 | 2.85 | 0.35 | -1.32 | 3.61 | 0.49 | 2017-03 | - | - | - | - | - | - | 2019-03 | - | - | - | - | - | - |
| 2015-04 | -1.11 | 2.13 | 0.50 | -1.48 | 2.52 | 0.00 | 2017-04 | - | - | - | - | - | - | 2019-04 | - | - | - | - | - | - |
| 2015-05 | -2.15 | 2.77 | 0.37 | -1.56 | 2.45 | 0.00 | 2017-05 | - | - | - | - | - | - | 2019-05 | - | - | - | - | - | - |
| 2015-06 | -2.12 | 2.63 | 0.46 | -2.29 | 3.26 | 0.83 | 2017-06 | - | - | - | - | - | - | 2019-06 | - | - | - | - | - | - |
| 2015-07 | -0.35 | 1.66 | 0.49 | -2.33 | 2.77 | 0.00 | 2017-07 | - | - | - | - | - | - | 2019-07 | - | - | - | - | - | - |
| 2015-08 | 1.44 | 2.55 | 0.26 | 0.92 | 2.84 | 0.87 | 2017-08 | - | - | - | - | - | - | 2019-08 | - | - | - | - | - | - |
| 2015-09 | 1.27 | 1.83 | 0.55 | 1.58 | 2.40 | 0.00 | 2017-09 | - | - | - | - | - | - | 2019-09 | - | - | - | - | - | - |
| 2015-10 | -0.81 | 2.04 | 0.28 | -0.90 | 2.04 | 0.00 | 2017-10 | - | - | - | - | - | - | 2019-10 | - | - | - | - | - | - |
| 2015-11 | -2.28 | 2.86 | 0.53 | 0.05 | 1.93 | 0.00 | 2017-11 | - | - | - | - | - | - | 2019-11 | - | - | - | - | - | - |
| 2015-12 | -1.50 | 2.77 | 0.50 | -3.33 | 4.34 | 0.00 | 2017-12 | - | - | - | - | - | - | 2019-12 | - | - | - | - | - | - |
| 2016-01 | -1.05 | 1.97 | 0.62 | -1.61 | 2.83 | 0.00 | 2018-01 | - | - | - | - | - | - | | | | | | | |
| 2016-02 | -1.60 | 2.99 | 0.24 | -1.04 | 3.49 | 0.04 | 2018-02 | -12.94 | 20.86 | 0.99 | -11.41 | 21.61 | 0.18 | | | | | | | |
| 2016-03 | -1.27 | 2.10 | 0.42 | -1.50 | 2.20 | 0.00 | 2018-03 | - | - | - | - | - | - | | | | | | | |
| 2016-04 | 0.37 | 1.58 | 0.53 | 0.91 | 2.21 | 0.00 | 2018-04 | -5.00 | 9.30 | 0.20 | -6.95 | 10.66 | 0.72 | | | | | | | |
| 2016-05 | -0.09 | 1.71 | 0.40 | -2.02 | 2.52 | 0.00 | 2018-05 | - | - | - | - | - | - | | | | | | | |
| 2016-06 | -2.12 | 2.87 | 0.16 | -4.22 | 4.99 | 0.64 | 2018-06 | - | - | - | - | - | - | | | | | | | |
| 2016-07 | -0.43 | 2.34 | 0.34 | -3.83 | 4.47 | 0.00 | 2018-07 | - | - | - | - | - | - | | | | | | | |
| 2016-08 | 1.10 | 2.44 | 0.21 | -2.25 | 3.59 | 0.02 | 2018-08 | - | - | - | - | - | - | | | | | | | |
| 2016-09 | 1.63 | 3.86 | 0.23 | 0.68 | 3.39 | 0.00 | 2018-09 | - | - | - | - | - | - | | | | | | | |
| 2016-10 | -0.04 | 1.53 | -0.51 | -0.26 | 2.00 | 0.00 | 2018-10 | - | - | - | - | - | - | | | | | | | |
| 2016-11 | -0.96 | 1.60 | 0.66 | 1.65 | 3.03 | 0.11 | 2018-11 | - | - | - | - | - | - | | | | | | | |
| 2016-12 | - | - | - | - | - | - | 2018-12 | - | - | - | - | - | - | | | | | | | |



**Table C4.** Analysis of the residual between CMAQ prior and posterior simulation and Burncluith site for 2015–2019. Averaged bias (Bias), Root-mean-square error (RMSE) and Pearson's coefficient (R).

| | | | | | | | Burncluith | | | | | | | | | | | | | | | |
|---|---|---|---|---|---|---|---|---|---|---|---|---|---|---|---|---|---|---|---|---|---|---|
| months | Prior | | | Posterior | | | months | Prior | | | Posterior | | | months | Prior | | | Posterior | | | | |
| yyyy-mm | Bias | RMSE | R | Bias | RMSE | R | yyyy-mm | Bias | RMSE | R | Bias | RMSE | R | yyyy-mm | Bias | RMSE | R | Bias | RMSE | R | | |
| 2015-01 | - | - | - | - | - | - | 2017-01 | -0.44 | 2.74 | 0.34 | 0.21 | 2.68 | 0.28 | 2019-01 | - | - | - | - | - | - | | |
| 2015-02 | - | - | - | - | - | - | 2017-02 | 0.368 | 1.77 | 0.34 | 1.39 | 1.97 | 0.68 | 2019-02 | - | - | - | - | - | - | | |
| 2015-03 | - | - | - | - | - | - | 2017-03 | -1.06 | 2.24 | 0.21 | 0.58 | 2.61 | 0.12 | 2019-03 | - | - | - | - | - | - | | |
| 2015-04 | - | - | - | - | - | - | 2017-04 | -0.77 | 2.02 | 0.36 | 0.31 | 1.60 | 0.65 | 2019-04 | - | - | - | - | - | - | | |
| 2015-05 | - | - | - | - | - | - | 2017-05 | -1.77 | 2.55 | 0.33 | -1.28 | 2.02 | 0.56 | 2019-05 | - | - | - | - | - | - | | |
| 2015-06 | - | - | - | - | - | - | 2017-06 | -0.7 | 1.82 | 0.16 | -1.79 | 2.49 | 0.14 | 2019-06 | - | - | - | - | - | - | | |
| 2015-07 | 0.86 | 2.12 | 0.41 | -1.03 | 2.33 | 0.29 | 2017-07 | -0.75 | 2.46 | 0.23 | -2.58 | 3.60 | -0.07 | 2019-07 | - | - | - | - | - | - | | |
| 2015-08 | 2.20 | 3.06 | 0.38 | 1.57 | 3.10 | 0.09 | 2017-08 | 1.133 | 1.79 | 0.25 | 0.90 | 1.54 | 0.47 | 2019-08 | - | - | - | - | - | - | | |
| 2015-09 | 2.03 | 2.69 | 0.44 | 2.08 | 3.16 | 0.27 | 2017-09 | 1.226 | 1.72 | 0.50 | 0.83 | 1.07 | 0.84 | 2019-09 | - | - | - | - | - | - | | |
| 2015-10 | 0.21 | 1.84 | 0.26 | 0.01 | 1.87 | 0.20 | 2017-10 | -1.24 | 3.07 | 0.48 | 0.94 | 2.44 | 0.72 | 2019-10 | - | - | - | - | - | - | | |
| 2015-11 | -1.24 | 2.23 | 0.73 | 1.21 | 2.30 | 0.69 | 2017-11 | -0.37 | 1.98 | -0.10 | 2.80 | 3.28 | 0.20 | 2019-11 | - | - | - | - | - | - | | |
| 2015-12 | 0.33 | 2.52 | 0.45 | -1.32 | 3.23 | 0.21 | 2017-12 | -1 | 1.94 | 0.28 | 0.41 | 1.51 | 0.50 | 2019-12 | - | - | - | - | - | - | | |
| 2016-01 | 0.92 | 2.78 | 0.46 | 0.01 | 3.30 | -0.11 | 2018-01 | -0.16 | 1.99 | 0.60 | -0.93 | 2.17 | 0.63 | | | | | | | | | |
| 2016-02 | -0.35 | 2.88 | 0.14 | 0.16 | 3.50 | -0.22 | 2018-02 | -2.31 | 3.11 | 0.38 | -2.47 | 3.40 | 0.17 | | | | | | | | | |
| 2016-03 | -0.50 | 2.06 | 0.58 | -0.58 | 2.04 | 0.59 | 2018-03 | -0.27 | 2.77 | 0.37 | -0.23 | 3.08 | 0.09 | | | | | | | | | |
| 2016-04 | 1.78 | 2.90 | 0.30 | 2.13 | 3.20 | 0.27 | 2018-04 | -0.32 | 1.84 | 0.49 | -1.58 | 2.11 | 0.51 | | | | | | | | | |
| 2016-05 | 0.92 | 2.05 | 0.30 | -1.04 | 2.05 | 0.14 | 2018-05 | -0.46 | 3.01 | 0.04 | -1.47 | 3.35 | -0.08 | | | | | | | | | |
| 2016-06 | -1.46 | 2.59 | 0.12 | -3.57 | 4.68 | -0.13 | 2018-06 | - | - | - | - | - | - | | | | | | | | | |
| 2016-07 | -0.482 | 1.99 | 0.38 | -3.90 | 4.45 | 0.40 | 2018-07 | - | - | - | - | - | - | | | | | | | | | |
| 2016-08 | 0.5448 | 2.52 | 0.07 | -2.79 | 4.26 | -0.02 | 2018-08 | - | - | - | - | - | - | | | | | | | | | |
| 2016-09 | 1.1873 | 2.70 | 0.58 | 0.36 | 2.37 | 0.62 | 2018-09 | - | - | - | - | - | - | | | | | | | | | |
| 2016-10 | 0.8141 | 1.79 | 0.48 | 0.43 | 2.06 | 0.13 | 2018-10 | - | - | - | - | - | - | | | | | | | | | |
| 2016-11 | -0.308 | 1.96 | 0.55 | 1.63 | 3.75 | -0.08 | 2018-11 | - | - | - | - | - | - | | | | | | | | | |
| 2016-12 | -0.548 | 1.34 | 0.78 | 0.62 | 1.44 | 0.77 | 2018-12 | - | - | - | - | - | - | | | | | | | | | |





## Appendix D: Prior, posterior and GPP flux anomaly correlation analysis

**Table D1.** Pearson's R correlations between prior and posterior climatological seasonal fluxes, and GPP fluxes derived from CABLE BIOS3, MODIS and DIFFUSE model.

| Bioclimate regions | Climatological seasonal cycle (2015-2019) | | | | | | |
|---|---|---|---|---|---|---|---|
| | Prior and Posterior | Prior and CABLE BIOS3 GPP | Prior and DIFUSSE GPP | Prior and MODIS | Post and CABLE BIOS3 GPP | Post and DIFFUSE GPP | Post and MODIS |
| Tropics | 0.73 | -0.66 | -0.50 | -0.51 | -0.46 | -0.32 | -0.33 |
| Savanna | 0.67 | -0.58 | -0.40 | -0.40 | -0.40 | -0.25 | -0.32 |
| Warm Temperate | 0.57 | 0.19 | 0.35 | 0.28 | 0.26 | 0.42 | 0.30 |
| Cool Temperate | 0.76 | -0.28 | -0.17 | -0.27 | -0.11 | -0.03 | -0.15 |
| Meditarranean | 0.83 | -0.27 | -0.15 | -0.19 | -0.30 | -0.19 | -0.26 |
| Sparsely vegetated | 0.33 | -0.23 | 0.01 | 0.00 | -0.21 | -0.12 | -0.30 |

**Table D2.** Pearson's R correlations between prior and posterior flux anomalies, and GPP anomalies derived from CABLE BIOS3, MODIS and DIFFUSE model.

| Bioclimate regions | Anomalies correlations ( 2015-2019) | | | | | | |
|---|---|---|---|---|---|---|---|
| | Prior and Posterior | Prior and CABLE BIOS3 GPP | Prior and DIFUSSE GPP | Prior and MODIS | Post and CABLE BIOS3 GPP | Post and DIFFUSE GPP | Post and MODIS |
| Tropics | 0.59 | -0.63 | -0.20 | -0.38 | -0.34 | -0.02 | -0.15 |
| Savanna | 0.59 | -0.73 | -0.61 | -0.63 | -0.50 | -0.45 | -0.38 |
| Warm Temperate | 0.43 | -0.64 | -0.52 | -0.52 | -0.32 | -0.32 | -0.29 |
| Cool Temperate | 0.20 | -0.65 | -0.50 | -0.52 | 0.10 | -0.03 | -0.09 |
| Meditarranean | 0.35 | -0.71 | -0.50 | -0.45 | -0.18 | -0.16 | -0.02 |
| Sparsely vegetated | 0.49 | -0.46 | -0.31 | -0.34 | -0.61 | -0.49 | -0.48 |





*Data availability.* Data available on request from the authors

*Code availability.* The code of the inversion system is available at https://github.com/steven-thomas/py4dvar (Thomas, 2020).

*Author contributions.* YV prepared all the input data required to run the inversion system and performed data analysis of the fluxes. YV was responsible for post-processing the TCCON and in-situ measurements, then developing the paper and figures. ST was the principal developer of the inversion system code. PJR and JDS also contributed to developing the inversion code, provided guidance for the manuscript's preparation and interpretation of the results. VH with help of JK ran CABLE BIOS3 and provided the biosphere fluxes required for the inversion. ZML provided data from the ground-based *in-situ* measurements (Cape Grim, Ironbark, Burncluith and Gunn Point) and gave comments on the paper. DP reviewed and comments on the TCCON Lauder site. ND and DG reviewed the final manuscript.

*Competing interests.* The authors declare that they have no conflict of interest.

*Acknowledgements.* This research was funded by the National Agency for Research and Development (ANID) scholarship, Becas Chile (grant no. 72170210) and supported by the Education Infrastructure Fund of the Australian Government, and the Australian Research Council (ARC) of the Centre of Excellence for Climate Extreme (CLEX, grant no. CE170100023). The authors would like to thank CSIRO and TCCON institutions for providing with the data to validate the inversion. Darwin and Wollongong TCCON stations are supported by ARC grants DP160100598, LE0668470, DP140101552, DP110103118 and DP0879468, and Darwin through NASA grants NAG5-12247 and NNG05-GD07G. NMD is funded by an ARC Future Fellowship, FT180100327. This research was undertaken with the assistance of resources and services from the National Computational Infrastructure (NCI), which is supported by the Australian Government, and the resources of the High-performance Computing Centre of the University of Melbourne, SPARTAN (Lafayette et al., 2016).The authors would like to thank all the OCO-2 MIP inverse modellers: Matthew Johnson, Frédéric Chevallier, Junjie Liu, Andrew Schuh, Andy Jacobson, Sean Crowell, David Baker, Sourish Basu and Feng Deng for contributing their OCO-2 (LNLG) global inversion products. The authors would also like to thank the institutions that provided data from the TCCON sites, and Vanessa Haverd from CSIRO in providing us with the Australian biosphere carbon flux data.



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
