# Peer review of "Interannual variability in the Australian carbon cycle over 2015-2019, based on assimilation of OCO-2 satellite data"

_Atmospheric Chemistry and Physics, 2022_

## Author Comment (AC1)

**Response to Referees' Comments**

Yohanna Villalobos Cortés

May 10, 2022

This document presents a point-by-point reply to the reviewers' comments on manuscript ACP-2022-15 (entitled "Interannual variability in the Australian carbon cycle over 2015-2019, based on assimilation of OCO-2 satellite data"). This reply is written on behalf of all co-authors.

We would like to thank to anonymous referees for their comments and efforts towards improving our manuscript. We took into account all the suggestions, and corrected the manuscript according to the reviewer's recommendations. The reviewer's comments are given in roman type, and our replies are shown in blue.

**Response to reviewer 1**

The manuscript by Villalobos et al., presents a regional inversion of  $CO_2$  fluxes over Australia, using OCO-2 observations and the CMAQ model. The study is well designed and the inverse modelling approach is sound and well described. Unfortunately, the paper lacks in at least two main aspects:

1. The presentation and discussion of the results lacks concision and depth: the authors produced many figures, which are analyzed one after the other, but there is no real effort of hierarchization of the conclusions from these analyses, and their cross-implications are not well explored.

We restructured the result section of the manuscript and provided a better description of the findings. As a result, we believe that the presentation of the findings throughout the current version of the manuscript is clearer and more concise than the previous version.

2. Specifically, the links between climate anomalies and  $CO_2$  flux anomalies are explored even before (and independently of) the robustness of the inversion results is assessed, which gives the impression that the authors try to fit their results in a pre-existing narrative, rather than verify if their results support it or not.

To better show our findings, we decided to change the order in which the results were presented. In the current version of the manuscript, the robustness of the inversion is discussed before the Australian carbon flux anomalies are explored. (Please see Sections 3.4 and 3.5 of the updated version of the manuscript).

Despite these negative points, the base for the study is sound and the paper can probably be improved significantly through major revisions of the text of some sections, but without the need for producing new simulations. I give more specific comments further below.

**Major comments**

The presentation of the results is very lengthy, but I find it poorly organized, and it lacks a hierarchization of the importance of the results and of their interpretations:

- The manuscript presents the results, then uses them to explore links between climate anomalies and CO2 flux anomalies, and only after that presents comparisons with independent data, and, at the very end, with results from other inversions. The scientific interpretation of the results is therefore done rather independently of their robustness assessment. Furthermore, this "robustness assessment" raises at least some suspicion on the results which, without invalidating them, makes it premature to jump into interpretations.
- There are many (17!) figures in these two sections, but a lot of repeated information from one to the other (e.g. Figures 6, 7, 8 and 16 all show prior/posterior flux anomalies + one additional indicator). Other figures are under-exploited (figures 10 and 11 show roughly the same thing), and information that needs to be interpreted in relation with each other end up on separate figures (Figures 13 and 14). Furthermore, the text is often just a very linear and lengthy description of the figures, and doesn't provide much added value (see specific comments below for examples): the text should guide the interpretation of the figures, not just describe them. This overall makes the paper rather hard to read, because the work of information filtering and hierarchization is largely left to the reader.

Figures 6, 7 and 8 were eliminated from the current version of the paper. We decided to replace those figures with Figure 9 (Please see Section 3.5 of the updated version of the manuscript). Fig. 9 shows that the prior/posterior flux anomalies, EVI, rainfall, and air temperature were combined into one plot. We decided to plot only the anomalies where that inversion is likely to have more accurate estimates of carbon fluxes (such as sparsely vegetated ecosystems and savanna.)

Regarding the content itself, there are also at least two major issues, that will need to be addressed in a revision:

• In comparisons with independent observations (surface-based and TCCON), the inversion tends to degrade the fit to independent data (especially at the surface sites). Some possible explanations are mentioned (retrieval biases due to clouds, possible transport model errors, non-representativity of the independent observations), but the implications of these on the interpretations of the results is not discussed. Furthermore, in comparisons with inversions from the OCO2 MIP project, the CMAQ inversion is quite an outlier, which should further raise at least some carefulness regarding the scientific conclusions that can be derived from these results. On a side note, I missed a discussion on the validity of the boundary condition (whose influence at the observation sites is not even shown in the figures). Given the poor fit to oceanic observations (1545: "all the negative large posterior biases [...] are associated with [...] winds that come from the ocean"), it could very well be a large source of systematic error.

We did not show the influence of the boundary condition at the observation site because the WRF-CMAQ domain where we performed the inversion has a large buffer around Australia.

In the current version of the manuscript, we incorporated the area of our study domain (please see Fig.1). From this figure, the reviewer will see that our study area covers the Australian continent and other countries such as Indonesia, Papua New Guinea, and New Zealand. The extension of this domain was made as an extra precaution to minimize the influence of the boundary conditions over Australia.

We showed in (Villalobos et al., 2021) that the main reason why our inversions degrade the fit with independent data is that we do not include OCO-2 ocean glint data. The reviewer can see in Villalobos et al. (2021) that adding ocean glint observation to our system improves that fit with surface measurements (Figure. 12 Villalobos et al., 2021). In this study, we decided not to use ocean glint observation because it has been shown that ocean glint observations in OCO-2 version 9 are subject to potential biases that could lead to misleading carbon flux estimates. In future work, we might use ocean glint data from OCO-2 version 10, which biases treatment is better than version 9.)

• A lot of focus is put in trying to highlight links between CO2 flux anomalies and anomalies in weather/climate parameters (temperature and precipitations) and other relevant products (EVI, GPP). However, the link is not that obvious. For instance, in five of the six ecosystems studied, the inversions leads to a reduction of the correlation between C flux and EVI (which could in fact be interpreted as the inversion refuting that link, somehow). Correlations between rainfall and C flux are ≤ 0.16 in five of the six ecosystems, and correlations with temperature are also not very convincing except maybe in the "Cool Temperate" and "Sparsely vegetated" regions. The authors are relatively careful regarding the wording of their conclusions, yet, this link between C flux and climate anomalies seems one of the focal points of the paper (e.g. last sentence of the abstract). I don't think that these poor correlations should necessarily be interpreted as a refutation of links between climate and flux anomalies, but maybe establishing this link requires more than just comparing time series: there's a reason why we need DGVMs!

We understand the concern of the reviewer. Most of the ecosystems that show a correlation reduction are ecosystems strongly influenced by ocean fluxes. Based on the robustness analysis with independent data, we decided it was better to assess the anomalies and their drivers over Australia's largest ecosystems, such as savanna and sparsely vegetated ecosystems.

DGVM models are valuable tools that help us understand global terrestrial carbon patterns; however, they are highly simplified representations of the real terrestrial biosphere. A reduction of correlation between fluxes and anomalies between the prior (a model) and posterior may well arise from complexities the model does not capture. In a recent Australian carbon cycle assessment made by (Teckentrup et al., 2021), it was found that the magnitude of inter-annual variability in the NBP by 13 DGVMs models (TRENDY version 8) is remarkably varied, which discrepancies are largely explained by the land cover fraction of the vegetation (see Figure 8 in Teckentrup et al., 2021).

**Specific comments**

• Section 3.1 is basically just a description of Figure 2: I don't need to read what I can already see in the figure, but I would need guidance on how to interpret it: the posterior biases are systematically positive: is that normal? What is causing that huge negative prior bias in November 2017?

We restructured the text in Section 3.1 to give more guidance to the reader as follows:

Line 297 - 307: As an indication of the overall inversion performance, the Australian mean prior bias for 2015 - 2019 was reduced from 0.23 to 0.06 ppm, and the RMSE was reduced from 0.90 to 0.76 ppm (Appendix A, Fig. A1).

While we see that inversion reduces the prior biases significantly, relative small positive systematical posterior biases remain (0.05 ppm). These systematic positive posterior biases across Australia may likely be driven by sampling and residual retrieval biases in the OCO-2 data. Some studies suggest that the existing OCO-2 cloud screening algorithm (Taylor et al., 2016) has difficulty identifying sub-field of view, and that unresolved clouds introduce a bias in the retrieved column of  $CO_2$  concentration.

We note the data gap in August and September was caused by a satellite outage. In November 2017, we saw the prior concentration underestimates the observations significantly, with biases of about -0.56 ppm and RMSE 1.29 ppm. High prior biases in this month were found along the east coast of Australia, suggesting that the CABLE model might likely be underestimating the carbon outgassing in this area and, therefore, the prior retrieval column  $CO_2$  concentration. The reduction of the prior biases in this month was about 90% (-0.06 ppm with an RMSE of 0.94).

• Section 3.2 is also just describing Figure 3. One question about this figure: how do the annual C budget compare between the prior and the posterior?

We restructured the whole text in Section 3.2. Note that Figure 3 was replaced by Figure 4 in the current version of the manuscript. Figure 4 includes the long term mean, annual and seasonal cycle of the prior and posterior carbon fluxes.

Figure. 4a shows the long term mean of the prior and posterior carbon fluxes aggregated across Australia for the period 2015 - 2019, and Figs.4b and c show the annual, and seasonal cycle of these estimates. Posterior flux uncertainties from 2016 to 2019 were assumed to be the same as those calculated for 2015, which were estimated by five different observing system simulation OSSE experiments (see more details in Villalobos et al., 2021).

Modified text: line 320 - 349. Our five year inversion suggests that Australia was a carbon sink of  $-0.46 \pm 0.09$  PgC yr-1 compared to the prior flux estimate, which was  $0.11 \pm 0.17$  PgC yr-1 (Fig. 4a). Here, the prior flux estimate (fluxes derived by the CABLE model) represents the current knowledge of the Australian carbon budget. Due to the size of the uncertainties in the prior estimate, it cannot be concluded with high confidence whether Australia was a sink or source of CO2 for the period 2015 – 2019. The annual posterior fluxes also suggest that Australia's terrestrial biosphere is able to absorb more carbon from the atmosphere than the CABLE model estimate (Fig.4b). We also see that 2016 was the year that largely contributed to the long term mean sink estimated by the OCO-2 inversion.

In terms of seasonal cycle, we can see that the posterior flux estimates show a stronger seasonality compared to the prior flux estimate (Fig. 4c). Over the five years from 2015 to 2019, we see that OCO-2 sees a strong seasonal biospheric carbon uptake each year between June and September (winter and early spring in Australia), and a stronger carbon source from November to December (late spring and early summer in Australia). As we showed in Fig. 3, the stronger carbon uptake seen in winter and early spring occurs because the prior column average concentration simulated by CMAQ model overestimate OCO-2 observations in this period.

To identify which regions the OCO-2 satellite sees a stronger carbon uptake in Australia, we plotted the annual map difference between the posterior and the prior fluxes (Fig. 5). We can see in Fig. 5a that the majority of the posterior long-term mean flux for the period 2015 to 2019 is distributed in one half of the continent (in the northeast, central and southern regions of the continent). However, we note that this was not the case for the coastal region in these areas, where we observe that OCO-2 recorded a stronger carbon release compared to the prior estimate.

The substantial difference between the prior and posterior flux in 2015 and 2016 comes from the northern and southeast of Australia (excluding coastal areas in the southeast of the continent). We will show later in Section 3.5 that the stronger carbon uptake recorded by the inversion (relative to the prior) in these two years was driven by an increase in vegetation productivity due to a rise in rainfall and low temperature across these regions. Despite the fact that 2016 was one of the strongest El Niño events on record in the Pacific Ocean, the rain over Australia was above average for most of the continent. The annual climate report from the Bureau of Meteorology for 2016 indicates that the annual rainfall over Australia was 17 per cent above the 1961–1990 average. In 2017, prior and posterior differences were seen in the northern, central and east coastal areas of Australia. Rainfall in 2017 was below average for much of eastern Australia and along the west coast of Australia. For 2019, OCO-2 recorded a stronger carbon release in western and central Australia. These results are not unexpected because 2019 was an exceptional year (the hottest and driest year on record in Australia), where the mean temperature was 1.52 °C above the 1961–1990 average (Annual climate statement, Bureau of Meteorology, 2019). We also noticed a substantially large carbon uptake (relative to prior) in the southeast corner of Australia recorded in 2019 (Fig. 5f).

• Section 3.3.1: How is that "Results"? And again, this is just a (long) description, subplot by subplot and year by year, of the information shown in Figure 6. But what's the take home message of that? Similarly, Section 3.3.2 basically just describes Figure 7 and 8.

As stated before, figures 6, 7 and 8 were eliminated from the current version of the paper. Instead, we replaced them with Figure 9 (Please see Section 3.5. in the updated version of the manuscript)

• Figures 6, 7 and 8 could easily be merged into one. This would also make it easier to see if

maybe there is a combined effect of e.g. temperature and precipitation.

We merged figures 6, 7 and 8 into one (Figure 9) (Please see Section 3.5. in the updated version of the manuscript).

• Figure 10 shows correlations of the fluxes with temperature and precipitations at the pixel scale. The color scheme of that figure is terrible, it's impossible to distinguish a correlation of 0.5 from a correlation of 0.8. Also, the figure doesn't say if the correlations have improved or degraded compared to the prior, which would be required for a proper interpretation.

Figure 10 was replaced by Figure 11 (Please see Section 3.5 in the updated version of the manuscript). Figure 11 shows spatial map of monthly temporal correlation between (a, b) EVI anomalies, prior anomalies and posterior anomalies. (c, d) rainfall anomalies, prior and posterior anomalies. (e, c) air temperature anomalies, prior and posterior anomalies for the period 2015–2019. The colour scheme of the plot was also changed.

• Section 3.4 and 3.5: Again, there's no need to describe the figures so extensively, I can see myself that the fit is sometimes improved, sometimes degraded. But is it expected? Does it help understanding what you showed in Figures 2 and 3? What implications does it has for the relationships between flux and climate parameters that you looked at in Section 3.3?

The assessment of the inversion (Section 3.4 and 3.5) was merged into one Section 3.4. In this new section of the manuscript we added the following text:

Line 450 - 459: A poor fit between the posterior concentrations and surface sites raises doubts about the reliability of the OCO-2 assimilated fluxes estimated over warm temperate, tropics, and cool temperate ecosystems. Therefore, in the upcoming section, we assess the analysis of the variability of the posterior fluxes only over the savanna and sparsely ecosystems, where our posterior carbon fluxes derived by OCO-2 data are likely more trustworthy than fluxes assimilated over areas directly impacted by off-shore ocean fluxes.

• Figures 10 and 12 are quite hard to read (too much information with the error bars). Figures 10 and 11 are maybe redundant (and same for 12 and 13).

Figures 10 and 11 were merged into one (Figure 7). Figures 12 and 13 were also merged (Figure 8). Please see Section 3.4 of the current version of the manuscript. Box plots were eliminated from the Figures.

• Section 3.5, 1518-524: "One possible explanation [...] vertical transport of the CMAQ model": why isn't this discussed more? This degradation of the fit to surface observations proves that at least something is going wrong in your inversion. Maybe it can be ignored, but then you need to justify this!

We modified some part of the text in this section of the manuscript, and provide more explanation of why our inversion might lead to large negative bias in Burcluith site (mainly in winter season).

Modified text: line 427 - 433: Large negative posterior biases at this site could be related to errors in the transport of the CMAQ model (e.g., associated with parameterization scheme

within the planetary boundary layer) or erroneous meteorological inputs from our WRF simulations (forcing errors). Transport errors in the vertical mixing near the surface associated with incorrect treatment of atmospheric turbulence can cause significant biases in simulated concentrations (Gerbig et al., 2008; Lauvaux et al., 2012). The atmospheric boundary layer mixing height is an important property in atmospheric modelling because it gives the volume of a column of air in which the fluxes contribute to the  $CO_2$  concentration. In this study, it is difficult to quantify the likely error in the simulation of boundary layer height because the site lacks the relevant physical measurements. More discussion of these findings is found in Section 4. We also did not find much improvement in the correlations at this site (see Appendix G, Table G4

• Figures 14 and 15 need to be merged. It's really difficult to jump from one page to the other to understand the link between the two, and they are not that useful on their own (well, Figure 14 is, but then it should have been shown much earlier).

We followed the reviewer's advice and decided to bring Figure 14 earlier in the text to connect the manuscript's ideas better. Figure 14 was replaced with Figure 6 in the current version of the manuscript. Please see Section 3.3.

• In Section 4, there are a lot of comparisons with GPP from other data products (the prior CABLE BIOS3, the DIFFUSE model, MODIS data) and with other inversions: this is useful, but again, it needs to be connected together and to the rest of the manuscript and to broader research questions. There are efforts in that direction, but it needs to be more refined. For now, it still comes up quite a bit as a long list of comparisons, analyzed one by one, rather than as pieces of a larger puzzle. What's are the scientific questions that the three subsections try to address? How robust are these discussions, given what has been seen in other parts of the paper?

In order to better discuss the findings in our manuscript, we decided the bring the GPP findings earlier in the document (Please see sections 3.3 and section 3.5). We updated Section 4 and added an extra dataset to compare our inversion (the ensemble mean from FLUXCOM product) as a suggestion of reviewer 2. Please see lines 499-571 in the updated version of the manuscript.

• Appendices: I think most of it is superfluous (what's the added value of showing 60 plots of spatial distribution of the OCO2 soundings vs. showing e.g. one example month or year?)

We eliminated some of the spatial distribution maps of OCO-2 from the appendix and only included the year 2015.

Citation: https://doi.org/10.5194/acp-2022-15-RC1

**Response Referee 2**

Villalobos et al. (2022) describes a regional inversion for Australia over the period from 2015 to 2019, expanding on work presented in Villalobos et al. 2020 and 2021. The inversion assimilates measurements from the OCO-2. The CMAQ atmospheric transport model was used to simulate transport and dispersion, driven by meteorological data from the WRF model. The CABLE model was used to provide estimates of biospheric terrestrial fluxes, forced with Australian regional drivers and observations from the BIOS3 set-up. Prior fossil fuel emissions were obtained from ODAIC, with missing sectors taken from EDGAR product, and diurnal factor from Nasser (2013). Ocean fluxes were obtained from the CAMS global model (Chevallier 2019), fire fluxes from GFED version 4.1. CAMS was also used to provide information on initial and boudary conditions.

Validation was carried out by comparing posterior concentrations to those measured at TCCON sites and ground-based in-situ measurements. Fluxes were compared to those from nine other atmospheric inversions and their ensemble.

To understand the association between bio-climatic factors and CO2 fluxes, rainfall was obtained from Australian Water Availability Project (AWAP), Bureau of Meteorology (BOM) and temperature from ERA5 ECMWF atmospheric reanalyses. EVI was used as a proxy for vegetation greenness and activity. Fluxes between the posterior fluxes and bio-climatic factors were assessed. Some evidence supported the hypothesis that the Australian savanna ecosystem during 2015/2016 period following higher than average rainfall was a carbon sink on average. But there is still a large amount of uncertainty in these estimates as the validation of the posterior fluxes against the available modelled biogenic fluxes and posterior concentrations compared with situ measurements did not improve greatly compared with the prior flux and concentration estimates.

I would support the publication of this paper, but suggest some additional discussion be added to the manuscript so that the reader can better understand why the comparison of modelled concentrations with in situ measurements is poor, and what would be required in order to obtain the validation of the inversion results for terrestrial Australia, which would then allow for more robust conclusions to be drawn regarding the source and sink status of biomes in Australia using the posterior fluxes from OCO-2 inversions.

Main Comments:

• The objective of the paper was to assess the interannual variability in CO2 fluxes in relation to bio-climatic factors. This is often the objective of many papers based on eddy-covariance site measurements, where fluxes are directly measured. There is an absence in the paper in discussion around how findings from eddy-covariance measurements taken during this period compares with fluxes estimated from OCO-2 inversion and other inversions, where fluxes are indirectly obtained. The paper would also be improved if some discussion were included on what is known about interannual variability of CO2 fluxes in other savanna ecosystems in response to bio-climatic factors. E.g. Williams et al. (2008), Archibald et al. (2009), and Merbold et al. (2011).

In the discussion section, we added information derived from FLUXCOM eddy-covariance-fluxbased product. We believe that a direct (point-by-point) comparison with the OzFlux network is not appropriate because of the sparseness of the flux tower sites across Australia. Besides, the flux tower site measurements have a relatively small footprint compared to the grid-cell scale where we performed the inversion (grid-cell scale of 81 km.). The reviewer can find some of this information in the following lines of the manuscript:

Line 519-535: We also studied the carbon flux anomalies derived by the OCO-2 MIP, FLUXCOM and compared them with the prior and posterior flux anomalies (3-month running mean) that we have discussed throughout this study (Fig. ??). We see in Fig. ?? that all carbon flux estimates agree that 2016 was the period that Australia recorded the largest carbon uptake relative to the 2015-2018 mean. We saw throughout this study that 2016 was a year that Australia recorded above-average precipitation and low temperatures that certainly drove the increase in vegetation productivity across the country. Similar findings were found by Haverd et al. (2016) in 2011, which results suggest that the variations of carbon fluxes over Australia's semi-arid ecosystems have a direct physiological response of vegetation productivity to water availability fluctuations. Other regional studies made in Africa (e.g., Williams et al., 2008; Archibald et al., 2009; Merbold et al., 2009), also indicate that interannual carbon fluctuations of semi-arid ecosystems largely depend on water availability driven by variations in rainfall between years. Water availability is the most important factor that controls the vegetation productivity of ecosystems across most of Australia, such as grassland and shrub/desert (see Figure 2 in Churkina and Running, 1998)

In terms of the amplitude of carbon flux anomalies, we can see that the prior and the FLUXCOM anomalies exhibit a lower amplitude than the one derived by our inversion and the majority of the models in MIP. Australia FLUXCOM estimates are likely not a good representation of the carbon flux estimates in the continent, given the sparsity of the flux tower network. FLUXCOM carbon fluxes use machine learning methods to empirically upscale flux tower data. In Australia, the number of OzFlux networks is small (approximately 30 towers), where most of the flux towers are located far away from semi-arid/arid ecosystems. This is relevant for Australia because semi-arid/arid ecosystems represent about 70% of the Australian land.

• As already discussed by the first reviewer, it would be helpful to understand how the boundary conditions contributed to the posterior solution, and what sort of magnitude of correction was made to these concentrations by the inversion. Another issue that is discussed by the authors is that during this period, Australia experienced some of the worst wild fires on record. Was there any special treatment applied to the OCO-2 measurements to filter out periods when the transport model would like have performed very poorly, such as during these fires?

In the method section we incorporated how we treated the boundary conditions in our inversion system. (Please see Section 2.2)

Line 96 - 104: To avoid the effect of initial conditions (ICs) and boundary conditions (BCs) on our OCO-2 assimilated carbon fluxes, we also optimized them within them the control vector  $\mathbf{x}$ . Each lateral boundary (south, east, north, and west) of our regional WRF-CMAQ domain was split into two regions. Lateral BCs at lower layer of the atmosphere were taken

from  $\sigma = 1$  to  $\sigma = 0.96$ , which correspond (on average) to a pressure of 972.5 hPa, while the upper boundary layer were solved from 972.5 up to 50 hPa. Each lateral BCs was solved at a monthly scale, but they were provided to our system as daily averages. Boundary and initial concentration were taken from CAMS global CO2 atmospheric inversion product data (version v19r1) (Chevallier, 2019). BCs uncertainties were assumed as the standard deviation (1 $\sigma$  uncertainty) in the perimeter of each region of the boundaries, and uncertainties for the initial condition were set at 1% (approximately 4 ppm). An diagram of the WRF-CMAQ domain is illustrated in Fig.1.

Regarding whether or not we applied any special treatment to OCO-2 measurements during the high fire event, we did not use any extra special treatment. We only selected "Good quality data" from the OCO-2 lite file. Bad OCO-2 soundings, e.g. those affected by aerosols from fires, are screened out by the A band preprocessor and IMAP DOAS preprocessor before the ACOS L2FP algorithm performs retrievals (Kiel et al., 2019).

• As shown in the results, for several of the biomes and periods the inversion did not improve on the prior concentrations, and in fact made the agreement between the in situ measurements and modelled concentrations worse. The authors discuss the challenge of validating the posterior estimates from the inversion, given that many of the in-situ sites were coastal sites where OCO-2 retrievals were not obtained. Given that it may be a large challenge to obtain reliable retrievals of OCO-2 fluxes over the ocean, what suggestions do the authors have to improve validation over continental Australia?

We believe that the validation against the in-situ measurements will improve significantly if we add to the inversion ocean glint observations from the current version of OCO-2 (version 10). OCO-2 algorithm team has confirmed the OCO-2 version 10 has reduced the biases and standard deviation compared to the TCCON data (OCO-2 Data Quality Statement, 2020). The analysis of how well the posterior concentration biases will improve using OCO-2 version 10 data will be assessed in future work. We added this information to the discussion section.

Line 554 - 566: More work needs to be done to reconcile and disentangle what is being found by the inversions and the Australia CABLE model. In future work, we could run this regional inversion using the latest version of OCO-2 data (version 10) in combination with ocean glint data, for which recent verifications confirm reductions in both the bias and standard deviation compared to the TCCON data (OCO-2 Data Quality Statement, 2020). Another direction for future work would be to explore the impact of transport model errors on the resulting assimilated OCO-2 fluxes. Such assessment could be done by choosing, for example, different planetary boundary schemes within the CMAQ model. As mentioned in section 3.4.2, a misrepresentation of vertical mixing near the surface in atmospheric transport models leads to uncertainties in modelled CO2 mixing ratios. Mixing within the planetary boundary layer influences the redistribution of the surface fluxes to the atmospheric column. Another way to evaluate the transport error of the model would be through a model inter-comparison. This approach is well-known in the global inversion TransCom group community (Law et al., 2008; Peylin et al., 2013; Basu et al., 2018), the recent model inter-comparison project (MIP) organized by the OCO-2 Science Team (Crowell et al., 2019; Peiro et al., 2021), and the recent European atmospheric transport inversion comparison (EUROCOM) project (Monteil et al., 2020).

Minor Comments:

Line 574: "constraint" should be "constrained"

**corrected**

Line 637: Duplication of "In summary"

**corrected**

References

Williams, C.A., Hanan, N.P., Baker, I., Collatz, J.G., Berry, J., Denning, A.S.: Interannual variability of photosynthesis across Africa and its attribution, Journal of Geophysical Research: Biogeosciences vol. 113 (G4), https://doi.org/10.1029/2008JG000718, 2008.

[revised manuscript text omitted]